# Lotus (*Nelumbo nucifera* Gaertn.) Leaf-Fermentation Supernatant Inhibits Adipogenesis in 3T3-L1 Preadipocytes and Suppresses Obesity in High-Fat Diet-Induced Obese Rats

**DOI:** 10.3390/nu14204348

**Published:** 2022-10-17

**Authors:** Yao He, Yue Tao, Liang Qiu, Wenfeng Xu, Xiaoli Huang, Hua Wei, Xueying Tao

**Affiliations:** 1State Key Laboratory of Food Science and Technology, Nanchang University, Nanchang 330047, China; 2Department of Medical Translational Center, Jiangxi University of Traditional Chinese Medicine, Nanchang 330004, China

**Keywords:** lotus leaf, *Enterococcus*, fermentation, adipogenesis, anti-obesity, dietary supplement, functional foods

## Abstract

The lotus (*Nelumbo nucifera* Gaertn.) leaf is a typical homologous ingredient of medicine and food with lipid-lowering and weight-loss effects. In the present study, lotus leaves were fermented by two probiotics, *Enterococcus faecium* WEFA23 and *Enterococcus hirae* WEHI01, and the anti-adipogenic effect of *Enterococcus* fermented lotus leaf supernatant (FLLS) was evaluated in 3T3-L1 preadipocytes with the aim of exploring whether its anti-obesity ability will be enhanced after fermentation with *Enterococcus* and to dig out the potential corresponding mechanism. The FLLS fermented by *E. hirae* WEHI01 (FLLS-WEHI01) was selected and further investigated for its ability to inhibit obesity in vivo in high-fat diet (HFD)-induced obese rats (male, 110 ± 5 g, 4 weeks old) due to its superior inhibitory effect on adipogenesis and lipid accumulation (inhibition rate of up to 56.17%) in 3T3-L1 cells (*p* = 0.008 for WEHI01-L, *p* < 0.001 for WEHI01-H). We found that the oral administration of both the low and high doses of FLLS-WEHI01 could achieve some effects, namely decreasing body weight (*p* < 0.001), epididymal fat mass, adipocyte cell size, LDL-C levels (*p* = 0.89, 0.02, respectively), liver TC levels (*p* < 0.001, *p* = 0.01, respectively), and TG levels (*p* = 0.2137, *p* = 0.0464, respectively), fasting blood glucose (*p* = 0.1585, *p* = 0.0009), and improved insulin resistance (*p* = 0.33, 0.01, respectively) in rats of the model group. Moreover, the administration of both high and low doses of FLLS-WEHI01 decreased the transcription levels of adipogenic transcription factors and corresponding genes such as *Pparγ* (*p* < 0.001), *Cebpα* (*p* < 0.001), *Acc* (*p* < 0.001), and *Fas* (*p* < 0.001) by at least three times. These results indicate that FLLS-WEHI01 can potentially be developed as an healthy, anti-obesity foodstuff.

## 1. Introduction

Obesity has proven to be one of the largest public health concerns worldwide because it is a fundamental contributing factor to many chronic metabolic diseases including type 2 diabetes mellitus (T2DM), hypertension, dyslipidemia, cardiovascular diseases, and different types of cancer [1]. Obesity is attributed to a disruption in the balance between energy absorption and expenditure; and cellular development is related to the growth of adipose tissue comprising both cellular hypertrophy—which leads to an increase in cell size—and hyperplasia—which leads to an increase in cell number—where cell hypertrophy is primarily determined by adipogenesis. The 3T3-L1 cell line is the known model used to investigate adipogenesis and the differentiation of preadipocytes [2]. Adipogenesis is a complex process modulated by a variety of transcription factors, such as the peroxisome proliferator activated receptor γ (PPAR γ), CCAA/enhancer binding protein (C/EBP) family members, and sterol regulatory element binding protein 1c (SREBP-1c); they also regulate the expression of downstream target genes involved in adipogenesis, such as acetyl CoA carboxylase (ACC) and fatty acid synthase (FAS) [3], which lead to morphological changes and lipid accumulation within cells. Multiple therapeutic methods with which to treat obesity (defined by body mass index and indicated where BMI > 30 (https://www.who.int/europe/news-room/fact-sheets/item/a-healthy-lifestyle---who-recommendations) (accessed on 6 May 2010) are accessible, such as lifestyle interventions, formula diets, drugs, and bariatric surgery. Concrete approaches used to reduce body mass vary from person to person and from purpose to purpose; the above strategies can be applied in combination or individually. Generally speaking, for pre-obese (BMI of 25–<30 kg/m^2^) people, it is suggested that they plan how to change their eating habits and do more exercise to achieve a permanent reduction of 5–10% in weight. The above modulation of nutrients and exercise is also applicable for grade I obesity (BMI 30–<35 kg/m^2^), but the use of a formula diet is more often recommended and refers to dietary foods used for special medical purposes, such as obesity treatment, and usually, industrially produced nutrients with a milk protein or soy protein base, which serve as complete or partial replacements of meals to reduce a parent’s initial weight. Moreover, drugs for reducing body weight such as orlistat, rimonabant, and sibutramine, are also an adjunctive measure for obesity therapy in the patients mentioned above. Further, when patients had a BMI > 35 kg/m^2^ and even >40 kg/m^2^, either multimodal conservative therapy (weight reduction programs) or bariatric surgery were suggested to reduce their body mass [4,5,6]. However, these therapies are often accompanied with negative side effects, such as poor-effect-induced repeated obesity as well as discomfort to other organs, such as the spleen, kidney, and stomach [7]. Recently, functional foods (such as natural plant products) have attracted much attention for their numerous beneficial effects, such as their anti-obesity effect and their replacement of, or coordination with, drugs [8,9,10]. For instance, tea, such as green tea, dark tea, and oolong tea, was proven to achieve an anti-obesity effect over a long period of time; Liu et al. further compared these and demonstrated that all of these modulated the gut flora in terms of both diversity and structure, which are closely associated with related indices of obese hosts via Spearman’s correlation analysis [11]. Additionally, radix Pueraria lobata (RP) was proven to inhibit the development of diet-induced obesity, the mechanism of which might be related to the enhancement of energy metabolism and the activation of PGC-1α as well as AMPK in the skeletal muscle [12]. Moreover, the anti-obesity effect of functional foods fermented with LAB has also received much attention. For example, the oral administration of *Lactobacillus plantarum*-BL2-fermented garlic extract reduced body weight and mass of epididymal-, retroperitoneal-, as well as mesenteric adipose tissue in HFD C57BL/6J male mice, on account of the fact that LAFGE can inhibit lipogenesis by down-regulating the transcription and translation levels of Pparγ, Cebpα, and lipogenic proteins [13].

*Nelumbo nucifera* Gaertn., commonly known as lotus, sacred lotus, Indian lotus, water lily, and Chinese water lily, is famous for its extensive usage as a dietary and medicinal plant that is widely distributed throughout East Asia. It has been a long time since the leaves, roots, seeds, and other parts have been used as food. As a traditional herbal medicine, the lotus, particularly the leaves, extract, fractions, and other constituents, has been proven to have diverse applications in both biology and pharmacology, i.e., antioxidant, antibacterial, antiviral, antifungal, immunomodulation, anti-inflammatory, anti-diarrheal, anti-thrombotic, anti-diabetic, cholesterol-lowering, anti-obesity, and anti-cancer properties [14,15,16,17,18,19,20,21], due to multiple bioactive compounds, including flavonoids, polyphenols, steroids, phenolic acids, polysaccharides, alkaloids, terpenoids, fatty acids, and glycosides [14,15,18,22,23]. In Asia, lotus leaves are usually applied as a functional food [24] and as a supplement with which to treat obesity. During the last several decades, many researchers have explored the probiotic characteristics of lotus leaves in regards to obesity [16,17,21,22,25,26], for these, some mechanisms of action are clear. To provide a specific example, Wang et al. reported that nuciferine (NUC) could inhibit the development of obesity, which might be accounted for by the fact that it can regulate the composition and potential capacity of intestinal microbiota, improve the function of the intestinal barrier, and prevent chronic low-grade inflammation [27]. In summary, the inhibitory effect of lotus leaves and their extract mainly impair the intestinal absorption of carbohydrates as well as lipids and increases energy expenditure, which may relieve chronic inflammation through ameliorating intestinal integrity and may modulate the gut microflora via specific metabolites such as short chain fatty acids. However, there are few reports on the anti-obesity effect of fermented lotus leaves, with only some researchers having demonstrated that bioactive substances such as the total phenol and flavone content, as well as a variety of low-molecular-weight metabolites, were increased after fermentation [28,29].

*Enterococcus* spp., belonging to LAB, are present in plenty of fermented foods and also within the guts of humans and animals [30] and have shown a wide range of beneficial properties [31,32,33], including an inhibitive effect on obesity [34,35]. Our previous studies have demonstrated that probiotics, *Enterococcus faecium* WEFA23 and *Enterococcus hirae* WEHI01, from healthy infants’ feces [36] were found to have the property of ameliorating obesity [37] and fermenting lotus leaves enhances their biological activities [38]. As such, we performed the fermentation of lotus leaves using *E. faecium* WEFA23 as well as *E. hirae* WEHI01 and examined the anti-obesity effect of them in vitro and in vivo, with the aim of demonstrating whether the anti-obesity capacity of lotus leaves will be enhanced after fermentation with *Enterococcus.* Thereafter, we dig out the potential and corresponding mechanisms by which to provide a foundation for the application of lotus leaves as a dietary complement to prevent and attenuate obesity.

## 2. Materials and Methods

### 2.1. Bacteria Strains and Culture Conditions

*Enterococcus faecium* WEFA23 and *Enterococcus hirae* WEHI01 were cultured in Brain Heart Infusion (BHI) broth (Oxoid, UK) anaerobicly at 37 °C.

### 2.2. Preparation of Enterococcus Fermented Lotus Leaf Supernatant (FLLS)

Cultures of *E. faecium* WEFA23 and *E. hirae* WEHI01 were centrifuged at conditions of 6000 rpm, 8 min, and 4 ℃ to harvest the bacterial pellets; afterwards, they were re-suspended (5 × 10^8^ CFU/mL) after being washed with PBS (Solarbio, Beijing, China) three times.

The fermentation of lotus leaf (LL) (lotus leaf was purchased from the Taobao application, and the link is https://m.tb.cn/h.UY2KvgW?tk=TBWz2wkdERJ CZ0001, and they grow naturally at the end of June and early July each year in the Weishan Lake, a fault lake located in the south of Weishan County, Jining City, Shandong Province, China) was performed by applying a traditional artisan method. Briefly, LL was crushed after drying and trimming, and was sieved through 80 mesh. Then, 2.5 g or 5 g LL of powder and 25 mL of distilled water were added into a conical flask prior to autoclaving at 121 °C for 15 min. Afterwards, 1 mL of a bacterial suspension of *E. faecium* WEFA23 or *E. hirae* WEHI01 was inoculated, followed by the adding of sterile distilled water to a total amount of 50 mL. LL without a bacterial inoculation was used as the control. Samples were anaerobically cultured at 37 °C for 24 h; *Enterococcus* fermented lotus leaf supernatant (FLLS; FLLS fermented by *E. faecium* WEFA23 and *E. hirae* WEHI01, named FLLS-WEFA23 and FLLS-WEHI01, respectively) and non-fermented lotus leaf supernatant (NFLLS) were obtained through centrifugation (6000 rpm at 4 °C for 20 min) and filtration. Finally, 6 samples were obtained, as follows: low concentration (50 mg/mL) of FLLS-WEFA23 and FLLS-WEHI01 (named WEFA23-L and WEHI01-L, respectively), high concentration (100 mg/mL) of FLLS-WEFA23 and FLLS-WEHI01 (named WEFA23-H and WEHI01-H, respectively), a low concentration (50 mg/mL) of NFLLS (named NFL-L), and a high concentration (100 mg/mL) of NFLLS (named NFL-H).

### 2.3. Cell Culture and Differentiation

The mouse 3T3-L1 preadipocytes were cultured in Dulbecco’s Modified Eagle Medium (DMEM; Solarbio, Beijing, China) supplemented with a 10% (*v*/*v*) fetal bovine serum (FBS, PAN, Edenbach, Germany) and 1% penicillin-streptomycin mixture of antibiotics (Solarbio, Beijing, China) in a humidified atmosphere of 5% CO_2_ at 37 °C.

Adipocyte differentiation was performed as described by Lee [39]. Briefly, 2.5 × 10^4^ 3T3-L1 preadipocyte cells were seeded into 24-well plates and incubated untill post-confluence for 48 h (defined as day 0 after 48 h post-confluence); they were then stimulated with fresh DMEM (with 10% FBS) and an adipocyte differentiation cocktail (MDI; 0.5 mm IBMX, 10 μg/mL insulin and 1 μM dexamethasone) (Solarbio, Beijing, China) for another 48 h, followed by incubation with DMEM (10% FBS and 10 μg/mL insulin). The induction was considered successful until the formation of lipid droplets was observed; during this period, DMEM containing 10% FBS was changed every 48 h.

### 2.4. Cell Viability Assay

The cytotoxic effect of *Enterococcus* fermented lotus leaf supernatant on 3T3-L1 preadipocytes was measuring with a Counting Kit-8 (CCK-8; Beyotime, Shanghai, China), as was the case in previous assays with some modifications [40]. The seeding of 5 × 10^4^ cells into 96-well plates was carried out, and then incubated for 24 h. Additionally, the cells were then treated with 1 μL of FLLS or NFLLS and incubated for 48 h, followed by another 1 h of incubation with new media containing 10 μL of CCK-8 solution. After incubation, the absorbance was measured at 450 nm and cell viability was calculated as follows:

The cell viability% = (OD (sample + cells)—OD medium)/(OD cells—OD medium) × 100.

### 2.5. Oil Red O Staining

The Oil Red O method is a well-known method by which to measure intracellular lipid accumulation [41]. Briefly, 3T3-L1 adipocytes undergoing differentiation were treated with 10 μL of FLLS or NFLLS on day 0, referred to in Section 2.3, after which the cells were stained, fixed, and observed by using a light microscope (Olympus, Tokyo, Japan). Finally, their absorbance was measured at 530 nm after being dissolved in isopropanol.

### 2.6. Animals and Experimental Design

Forty-eight male Sprague Dawley (SD) rats (110 ± 5 g, 4 weeks old) were housed in groups (eight rats were housed in a big cage that allowed them to move, and there was a total of six cages) in a standard plastic cage with a 12 h light/dark cycle and a controlled relative temperature of 23 ± 2 °C, in addition to the controlled relative humidity of 45–65%. The protocols of animal experiments in this paper were approved by the Animal Care Review Committee of the Nanchang University School of Medicine (Approval No. 0064257). The animal diagram was designed as follows: after acclimation for about seven days (rats were given the normal standard food during this period): (1) normal diet (ND): rats were given a normal diet and 0.9% saline; (2) high-fat diet (HFD): rats were given a high-fat diet, consisting of 66.5% basic feed, 20% sucrose, 10% lard, 2.5% cholesterol, and 1% cholate (Hunan slake Jingda experimental animal Co., Ltd.), as well as 0.9% saline; (3) NFL-L: rats were given the HFD and 1 mL of 50 mg/mL NFLLS (50 mg/kg BW/day); (4) NFL-H: rats were given the HFD and 1 mL of 100 mg/mL NFLLS (100 mg/kg BW/day); (5) WEHI01-L: rats were given the HFD and 1 mL of 50 mg/mL FLLS-WEHI01 (50 mg/kg BW/day); and (6) WEHI01-H: rats were given the HFD and 1 mL of 100 mg/mL FLLS-WEHI01 (100 mg/kg BW/day). Rats were given their corresponding diets for 5 weeks. The weights of the rats were recorded every day during this period.

### 2.7. Biochemical Analysis

The oral glucose tolerance test (OGTT) was performed after a 30-day intervention. After 12 h’s fasting, rats were orally gavaged with 2 g/kg BW of glucose, and caudal veins were used to collect blood samples at times of 0, 15, 30, 60, 90, and 120 min. Blood glucose was measured using a Sinocare Safe AQ Smart Blood Glucose Meter (Sinocare, Changsha, China). On the last day, the fasting glucose content was measured using an enzyme kit (Nanjing Jiancheng, Nanjing, China), and fasting plasma insulin was analyzed using a rat insulin ELISA kit (Nanjing Jiancheng, Nanjing, China). The homeostasis model assessment of insulin resistance (HOMA-IR) was calculated as described by Matthews [42]. Serum TC, TG, LDL-C, and HDL-C levels were detected using assay kits (Nanjing Jiancheng, Nanjing, China) according to the manufacturer’s instructions.

### 2.8. Hepatic TC and TG

Liver tissue was homogenized into a mixture (0.5 g of liver tissue in 4.5 mL of absolute ethanol). Liver TC and TG levels were detected via the method described above.

### 2.9. Histopathology Examination

The analysis of the histopathology of the rats’ tissues (liver and epididymal adipose) was carried out as previous described with some modifications [43]. For Oil Red O staining, there only existed differences in the dyestuff, and the procedure remained the same.

### 2.10. RT-qPCR

The total RNA from 3T3-L1 cells and epididymal adipose tissue was extracted using the TRIzol-Reagent (TaKaRa, Dalian, China) according to the manufacturer’s instructions. Reverse transcription and quantitative real-time PCR were performed in the same manner as that described in our previous experiment [38]. The primers used are listed in Appendix A.

### 2.11. Statistical Analysis

The data and the normal distribution of the data (the *p* of a Shapiro–Wilk normality greater than 0.05 were thought to conform to normal distribution) were analyzed using GraphPad Prism 7 statistical software. The results were expressed as the mean ± standard deviation (SD). All of the data passed the verification of normal distribution and were analyzed via a one-way analysis of variance and Tukey’s multiple comparisons test, used for comparisons between-groups, while the others were analyzed via non-parametric test. The *p*-values of less than 0.05 were considered statistically significant.

## 3. Results

### 3.1. Effect of FLLS on the Viability of 3T3-L1 Preadipocytes

Whether fermented lotus leaf supernatant (FLLS) exhibited any toxicity on 3T3-L1 preadipocytes was the first object of study. WEFA23-L, WEFA23-H, WEHI01-L, WEHI01-H, NFL-L, and NFL-H exhibited no significant effect on the cell viability of cells compared with the control, as presented in Figure 1, suggesting that both fermented lotus leaf supernatant FLLS and non-fermented lotus leaf supernatant (NFLLS) are not cytotoxic to cells.

### 3.2. Effect of FLLS on Intracellular Lipid Accumulation in 3T3-L1 Preadipocytes

To investigate the influence of FLLS on preadipocyte differentiation, the differences in lipid droplet formation induced by the adipocyte differentiation cocktail were compared. As shown in Figure 2, cells treated with FLLS showed a significant decrease in lipid content along with doses regarding the control group, in which the WEFA23-L, WEFA23-H, WEHI01-L, and WEHI01-H groups were reduced to 79.02% (*p* = 0.04), 56.57% (*p* < 0.001), 73.57% (*p* = 0.008), and 56.17% (*p* < 0.001), respectively. However, NFLLS showed no significant effect on the intracellular lipid accumulation compared with the control. These results demonstrated that FLLS fermented by both *E. faecium* WEFA23 and *E. hirae* WEHI01 could effectively inhibit lipid accumulation in cells.

### 3.3. Effects of FLLS on the Adipogenesis of 3T3-L1 Preadipocytes

To analyze the effects of FLLS on the adipogenesis of 3T3-L1, the transcription levels of C/EBP family members, such as C/EBPα and C/EBPβ, and their downstream target genes, i.e., ACC and FAS, were investigated by RT-qPCR. As shown in Figure 3a,b, the mRNA level of *Cebpα* significantly decreased after treatment with WEHI01-H (*p* = 0.0166). Similarly, the *Cebpβ* mRNA level also significantly decreased after being treated by WEFA23-H (*p* < 0.001), WEHI01-L (*p* = 0.023), and WEHI01-H (*p* = 0.0195). For lipogenic genes (Figure 3c,d), WEFA23-L (*p* = 0.005), WEFA23-H (*p* = 0.008), and WEHI01-H (*p* < 0.001) significantly decreased the mRNA level of *Fas*, while only WEFA23-H (*p* = 0.01) and WEHI01-L (*p* = 0.004) showed a significant inhibitory effect on *Acc* mRNA expression. However, NFL-L and NFL-H showed no significant effect on the mRNA expression of the test adipogenic genes compared with the control. Taken together, these results suggested that both FLLS-WEFA23 and FLLS-WEHI01 suppressed effects on adipogenesis and lipid accumulation in cells, with FLLS-WEHI01 having stronger impact, and therefore, selected for the investigation of its lipid-lowering effect in vivo.

### 3.4. Effect of FLLS-WEHI01 on Body Weight in Obese Rats

In order to determine the corresponding effect of FLLS-WEHI01 in vivo, an HFD-induced obese rat model was established. As shown in Figure 4a, during the feeding period, rats in the model group were significantly heavier compared to the control group. In contrast with the model group, the NFL-H (*p* = 0.03), WEHI01-L (*p* = 0.02), and WEHI01-H (*p* < 0.001) groups achieved a significant decrease in body weight in as early as the second week, while NFL-L achieved a significant decrease from the next week (*p* < 0.001). Till the end of this period, treatments with both NFL-L and WEHI01-H decreased the body weight of rats (*p* < 0.05) relevant to the model group (Figure 4b). Moreover, HFD feed significantly increased the liver weight (*p* < 0.001) and epididymis fat mass (*p* = 0.0104) in regard to the ND group, while the administration of WEHI01-L (*p* = 0.0331) and WEHI01-H led to a significant decrease in liver weight (*p* = 0.0338) (Table 1) and, only the latter decreased epididymal fat mass (*p* = 0.028) (Table 1) in rats fed the HFD diet. While both administration of WEHI01-L and WEHI01-H had no effect on the weight of the spleen and kidney as well as the food intake of rats.

### 3.5. Effect of FLLS-WEHI01 on the Glucose and Insulin Level in Obese Rats

To analyze the influence of FLLS-WEHI01 on glucose and insulin in obese rats, the OGTT and HOMA-IR in all groups were measured. As shown in Figure 5, the glucose tolerance of rats was improved significantly by the WEHI01-L treatment compared with HFD treatment (*p* = 0.03) (Figure 5a), and the fasting blood glucose was improved dramatically with a value of 8.11 mM by WEHI01-H (*p* = 0.0179) (Figure 5b). As for fasting insulin, there was no significant difference treatments with an ND or HFD (Figure 5c). However, the HOMA-IR index was significantly increased in the model group (*p* < 0.001), and only the administration of WEHI01-H effectively reduced the level, from a value of 25.31 to 15.19 (*p* = 0.001) (Figure 5d).

### 3.6. Effects of FLLS-WEHI01 on the Lipid Levels of the Serum and Liver in Obese Rats

The levels of biomedical substances in the serum and liver were determined at 5 weeks of administration as presented in Figure 6. The contents of serum TC (*p* = 0.0352), TG (*p* = 0.0375), LDL-C (*p* < 0.001), liver TC (*p* < 0.001), and TG (*p* = 0.0005) were significantly higher when rats were treated with an HFD compared to normal diet, while the serum HDL-C level was significantly converse (*p* < 0.001). On the contrary, the administration of WEHI01-H significantly decreased the levels of LDL-C (*p* = 0.02), liver TC (*p* = 0.01), and TG (*p* = 0.0464) in rats fed an HFD diet. However, WEHI01-L administration could only significantly decrease liver TC (*p* < 0.001) levels and achieve a reduced trend, similar to the normal group in serum and liver TG levels. However, the administration of NFL-L and NFL-H could only significantly reduce the levels of serum TG (*p* < 0.001).

### 3.7. Effects of FLLS-WEHI01 on the Histopathology of Liver and Epididymis Tissue in Obese Rats

After 5 weeks of oral administration, morphological analyses of the rats’ livers via H&E staining showed that hepatocytes in the HFD group became larger, with severe fatty lesions and the accumulation of lipid droplets in the cytoplasm (Figure 7a). Consistently, staining with Oil Red O showed that hepatocytes in the HFD group were full of lipid droplets (Figure 7b). The administration of NFL-L, NFL-H, WEHI01-L, and WEHI01-H could alleviate the HFD-induced hepatic accumulation of lipid droplets, injury, and doses. Among them, the WEHI01-H group had the best effect on reducing steatosis and hepatic lipid accumulation. In addition, analyses of rats’ epididymal adipose tissues via the H&E staining showed that adipocyte hypertrophy in the HFD group, that is, the sizes of adipocytes, was appreciably larger in the model group than in the control group. While the NFL-L, NFL-H, WEHI01-L, and WEHI01-H treatment considerably decreased the sizes of adipocytes compared with the HFD group (Figure 7c), these results suggested that the weight gain in obese rats is the result of adipocytic fat accumulation and the expansion of adipose tissue, and further, the administration of FLLS-WEHI01 and NFLLS could ameliorate the HFD-induced adipocytic hypertrophy, in which FLLS-WEHI01 exhibited a stronger effect.

### 3.8. Effect of FLLS-WEHI01 on the Expression of Adipogenic Genes in Epididymal Adipose Tissue

To explore the deep effect of the lipid-lowering effects of FLLS-WEHI01, the transcription expression levels of *Pparγ*, *Cebpα*, *Fas,* and *Acc* in the epididymal adipose tissue were examined by RT-qPCR. As shown in Figure 8, after 5 weeks of the administration of NFL-L, NFL-H, WEHI01-L, and WEHI01-H, the mRNA levels of *Cebpα*, *Pparγ*, *Fas,* and *Acc* in adipose tissue were significantly decreased regarding the HFD group (*p* < 0.001), which indicates that FLLS-WEHI01 and NFLLS ameliorated the HFD-induced obesity by inhibiting adipogenesis and lipogenesis in obese rats.

## 4. Discussion

The lotus has a history of more than 1000 years as a traditional herb. In recent decades, the lotus has attracted much attention from the scientific circle, and in-depth research has been carried out to evaluate the biological and pharmacological activities of multiple ingredients. Therein, the lotus leaf proved to have diverse biomedical applications, including anti-obesity, due to their numerous bioactive compounds. However, little was known about the function of lotus leaf after fermentation. Our previous study [38] showed that the fermentation of lotus leaf using *E. faecium* WEFA23 and *E. hirae* WEHI01 could enhance its antibacterial and antioxidant activities. This being the case, we performed the fermentation of lotus leaf by *E. faecium* WEFA23 and *E. hirae* WEHI01 and evaluated the anti-obesity effect of *Enterococcus*-fermented lotus leaf supernatant (FLLS).

We analyzed the effect of experimental samples on cell viability by using a CCK-8 kit, which was carried out to ensure the reliability of the following assay. The results of this showed that all of the experimental groups were not toxic to cells compared to the control. Contrary to our findings, Choi et al. demonstrated that the viability of 3T3-L1 preadipocyte cells was reduced when treated with a 1000 µg/mL extract of fermented white sword bean by *Bacillus subtilis* [44], suggesting that lotus was safer and more applicable. Furthermore, FLLS significantly reduced the lipid concentration (*p* < 0.05) and inhibited differentiation and adipogenesis in vitro, along with the dose level, while NFLLS showed no significant effect. Similarly, So KH et al. showed that fermented soybean FS extract (50 µg/mL) achieved a better inhibition effect on both the differentiation of adipocytes and the accumulation of fat during the differentiation of 3T3-L1 cells than nonfermented soybean (NFS) extract [45]. These results indicated that fermentation with lactic acid bacteria could be beneficial to the probiotic characteristics of anti-obesity, which might be because fermentation contributed to the release of bio-active substances. Interestingly, we measured the corresponding substances before and after fermentation with *Enterococcus*, and the results showed that the number of both crude polysaccharides and short chain acids increased after fermentation (data were not shown), which might be part of the reason why FLLS performed better than NFLLS.

Adipogenesis is a complicated process that involves many transcription factors, enzymes, and proteins, such as the key regulators of preadipocyte differentiation, such as ACC and FAS, which are pivotal in the synthesis of fatty acids, such as the synthesis of triglycerides and phospholipids [46], as well as acetyl coenzyme a and malonyl coenzyme a synthesis, into long-chain fatty acids [47], respectively. Therefore, the modulation of corresponding genes might be another reason. Our in vitro results demonstrate that FLLS could significantly down-regulate the mRNA levels of *Cebpα*, *Cebpβ, Acc*, and *Fas* in 3T3-L1 cells, which were consistent with Hwang’s study, which showed that *Bacillus subtilis* fermented soy bean extract inhibited lipid accumulation in 3T3-L1 cells through decreasing C/EBPα expression [48]. In addition, Kim demonstrated that fermented *Laminaria japonica* inhibited 3T3 cell adipocyte differentiation, due to the inhibition of the mRNA expression of *C/EBP-α/β* and *PPAR-γ* [49]. Moreover, it has been reported that fermented *Platycodon grandiflorum* (FPG) inhibited lipid accumulation in 3T3-L1 adipocytes via the regulation of PPARγ, C/EBPα, as well as fatty acid binding protein 4 (FABP4) [50]. Referring to the literature, we deduce that future research on apodosis biomarkers at both the mRNA and protein levels may better clarify the mechanism of action, which is a limitation of our research.

To further understand the potential mechanisms of the inhibitory effect of FLLS in vivo, an HFD-induced obese rat model was established, and FLLS-WEHI01 was selected for evaluation based on its stronger adipogenesis inhibition effect. Our results showed that the administration of a high concentration (100 mg/mL) of FLLS-WEHI01 (WEHI01-H) could significantly decrease weight gain, as well as epididymal fat mass, liver weight, and the hypertrophy of white fat in the epididymis in the HFD-induced obese rats, suggesting that WEHI01-H administration could effectively suppress fat accumulation, which is consistent with the results in vitro. As is well-known, obesity causes adipocyte proliferation, and hypertrophy, as well as an increase in the metabolism of adipocytes, thereby reducing the insulin sensitivity of skeletal muscle and producing insulin resistance, and thus, inducing type two diabetes [51]. The disordering of lipid metabolism, such as decreasing insulin resistance or increasing the release of adipokines, will further increase free fatty acids, which results in increasing fatty acids in the liver and TG accumulation in hepatocytes [52]. On the other hand, insulin resistance and elevated serum TG as well as cholesterol levels are often accompanied by hepatic steatosis, which leads to lipid metabolism disorders [53,54]. Based on our results, we found that the administration of FLLS-WEHI01 could effectively improve hyperglycemia and the insulin resistance of obese rats, as well as reduce the levels of biochemical substances in the serum and liver of obese rats. Furthermore, we found that WEHI01-H could alleviate liver injury, liver cell fibrosis, and lipid accumulation in hepatocytes through staining. Combined, these results suggest that the administration of WEHI01-H could effectively reduce blood glucose and regulate blood lipid disorder. Similarly, the fermentation broth of a Gegen Qinlian decoction reduced the levels of biochemical substances in the serum of T2DM rats and regulated abnormal glucose as well as lipid metabolism [55]. Hu et al. found that LAB-fermented Moringa leaf reduced liver lipid accumulation and inhibited inflammation in obese mice [56]. Moreover, our in vivo analysis of the modulation of relative genes suggested that WEHI01-H could alleviate hyperlipidemia, hepatic steatosis, and lipid metabolism disorders, which is consistent with previous reports that *Lactobacillus plantarum* BL2-fermented garlic extract significantly reduced the serum TG and TC levels of obese mice and that the mRNA level as well as protein levels of related genes were all down regulated [13].

Besides the above, there still remain many scientific problems concerning the probiotic effects of FLLS-WEHI01 on obesity which need to be further studied for elucidation. Firstly, the effect of lotus leaf-fermented supernatant on obesity explored in this study was only examined at the phenomenon level; therefore, more experiments are needed to further explore the specific mechanisms. Secondly, we used SD rats as the model organisms with which to explore the functional characteristic of FLLS-WEHI01, which indeed could not fully reflect the effect on the human body. Therefore, further studies are needed to elucidate the biological effects of FLLS-WEHI01 on obesity-related disorders and to clarify its effect on the health of human.

## 5. Conclusions

Figure 9 shows a schematic image of fermented lotus leaf treatment for inhibiting obesity in HFD-induced obese rat.The oral administration of FLLS-WEHI01 significantly decreased body weight, epididymal fat mass, TC and TG in liver, LDL-C levels, and fasting blood glucose, in addition to improving insulin resistance in HFD-induced obese rats, which might have occurred through the modulation of adipogenic transcription factors, such as *Pparγ*, *Cebpα*, *Acc,* and *Fas*. Therefore, the results suggested that FLLS-WEHI01 could potentially be applied as a dietary supplement for the prevention and attenuation of obesity.

## Figures and Tables

**Figure 1 nutrients-14-04348-f001:**
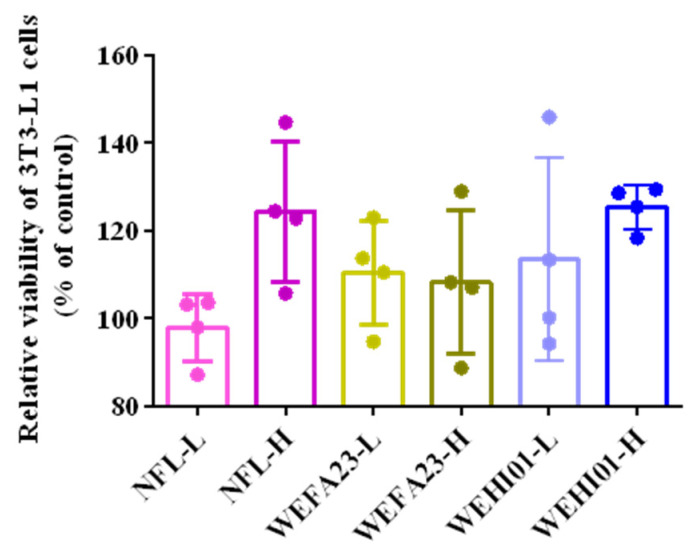
Effects of FLLS-WEFA23 and FLLS-WEHI01 on 3T3-L1 preadipocyte viability. Cells were treated with the *Enterococcus* fermented lotus leaf supernatant (FLLS) and non-fermented lotus leaf supernatant (NFLLS). Values are expressed as mean ± SD (*n* = 4).

**Figure 2 nutrients-14-04348-f002:**
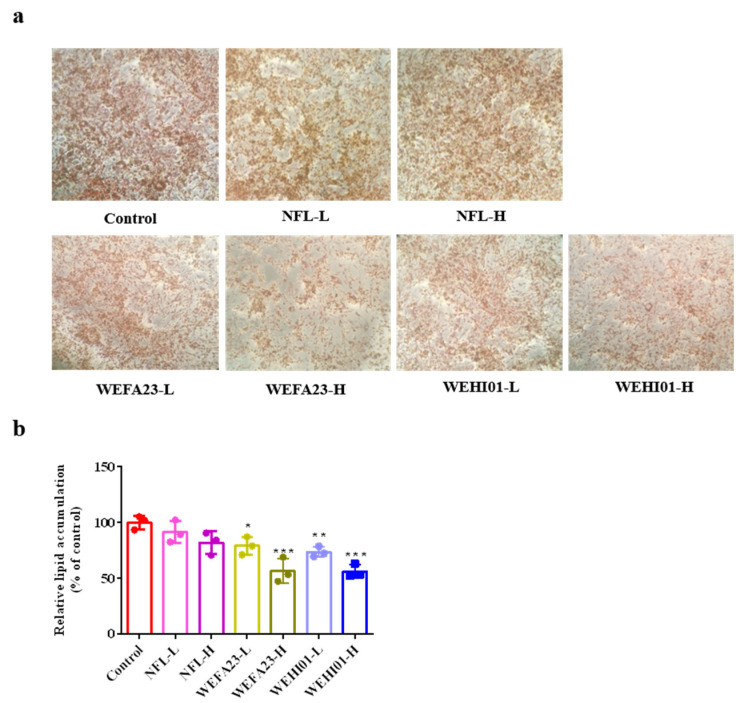
Effect of FLLS-WEFA23 and FLLS-WEHI01 on intracellular lipid accumulation in 3T3-L1 preadipocytes. (**a**) Representative images of lipid droplets which were measured by Oil Red O staining. Cells were treated with the *Enterococcus*-fermented lotus leaf supernatant (FLLS) and non-fermented lotus leaf supernatant (NFLLS) (10×). (**b**) Quantification of relative lipid content expressed as a percentage. Values were expressed as mean ± SD (*n* = 3). * *p* < 0.05, ** *p* < 0.01, and *** *p* < 0.001 vs. Control.

**Figure 3 nutrients-14-04348-f003:**
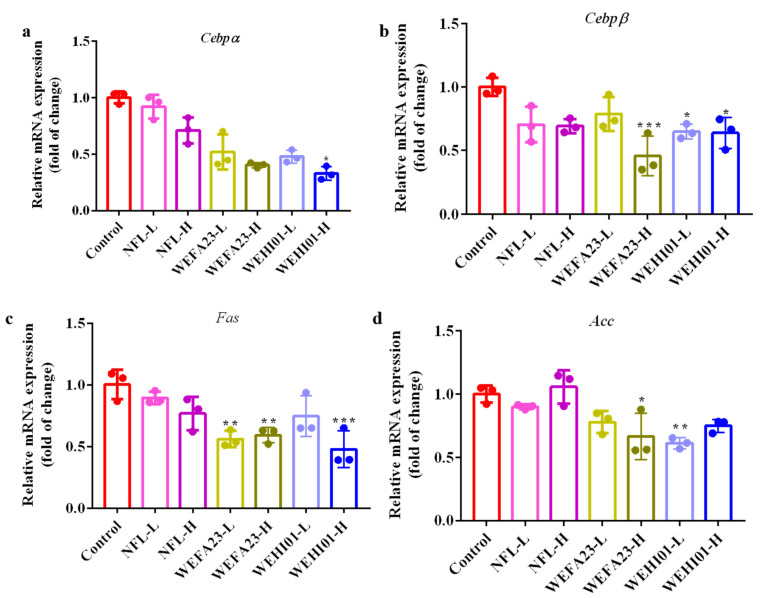
Effect of FLLS-WEFA23 and FLLS-WEHI01 on the mRNA expression of adipogenic genes in 3T3-L1 preadipocytes. The mRNA expression levels of (**a**) *Cebpα*. (**b**) *Cebpβ*. (**c**) *Fas*. (**d**) *Acc*. *GAPDH* gene was used as reference gene. Values were expressed as mean ± SD (*n* = 3). * *p* < 0.05, ** *p* < 0.01, and *** *p* < 0.001 vs. Control.

**Figure 4 nutrients-14-04348-f004:**
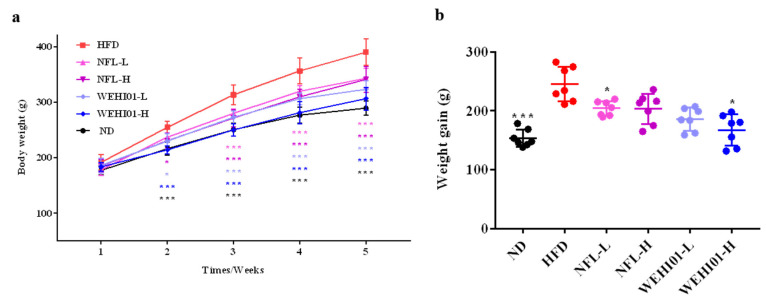
Effect of FLLS-WEHI01 on body weight in obese rats. (**a**) Body weight. (**b**) Body weight gain after 5 w. * *p* < 0.05, and *** *p* < 0.001 vs. HFD group.

**Figure 5 nutrients-14-04348-f005:**
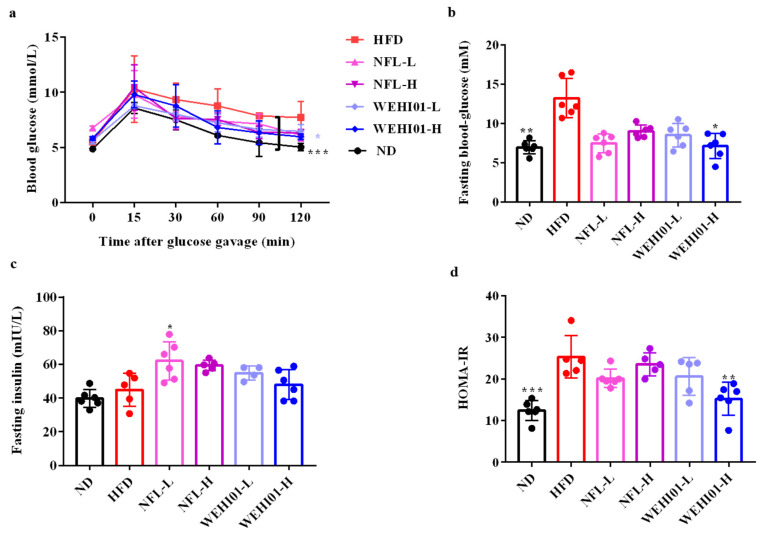
Effect of FLLS-WEHI01 on blood glucose levels and insulin resistance in obese rats. (**a**) Curve of the oral glucose tolerance tests. (**b**) Fasting blood-glucose. (**c**) Fasting insulin. (**d**) Homeostasis model assessment insulin resistance (HOMA-IR), calculated by fasting glucose content (mmol/L) × Fasting insulin content (mIU/L)/22.5. Values were expressed as mean ± SD (*n* = 6). * *p* < 0.05, ** *p* < 0.01, and *** *p* < 0.001 vs. HFD group.

**Figure 6 nutrients-14-04348-f006:**
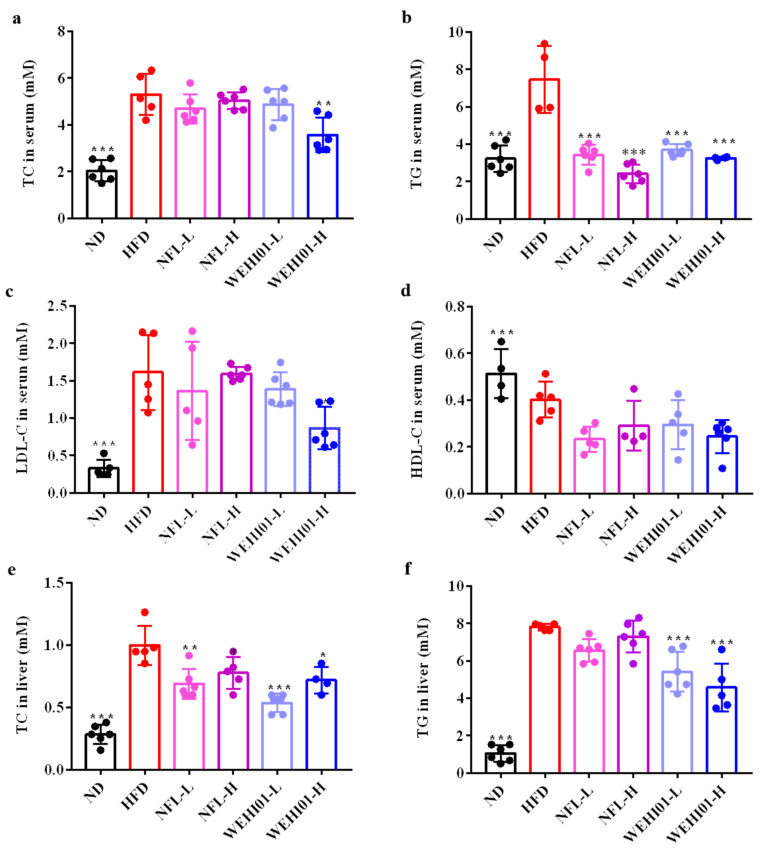
Effect of FLLS-WEHI01 on lipid levels of serum and liver in obese rats. (**a**) TC level in serum. (**b**) TG level in serum. (**c**) LDL-C level in serum. (**d**) HDL-C level in serum. (**e**) TC level in liver. (**f**) TG level in liver. Values were expressed as mean ± SD (*n* = 6). * *p* < 0.05, ** *p* < 0.01, and *** *p* < 0.001 vs. HFD group.

**Figure 7 nutrients-14-04348-f007:**
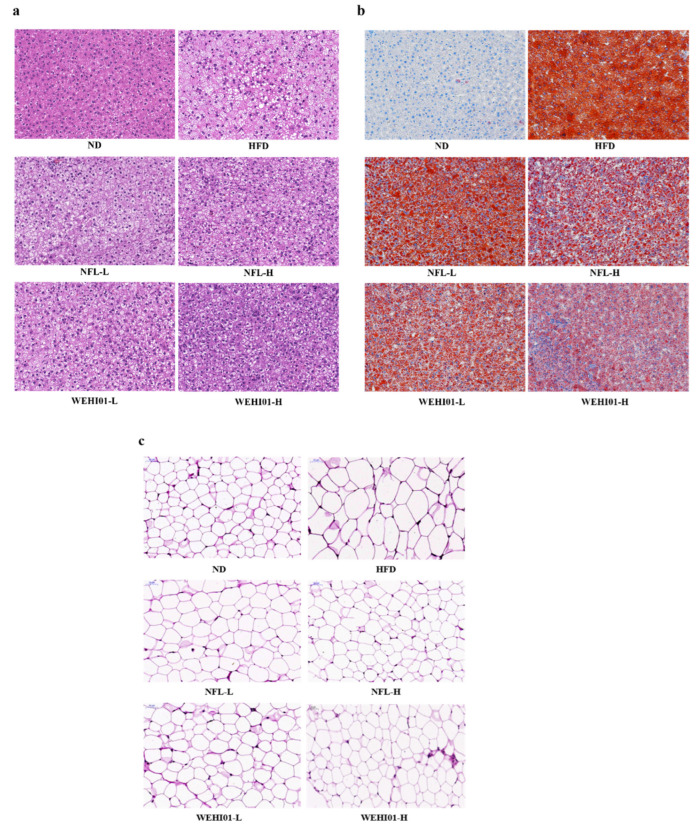
Effect of FLLS-WEHI01 on liver and epididymal tissue in obese rats. (**a**) H&E staining of liver. (**b**) Oil Red O staining of liver. (**c**) H&E staining of white adipose tissue of epididymis. Values were expressed as mean ± SD (*n* = 3).

**Figure 8 nutrients-14-04348-f008:**
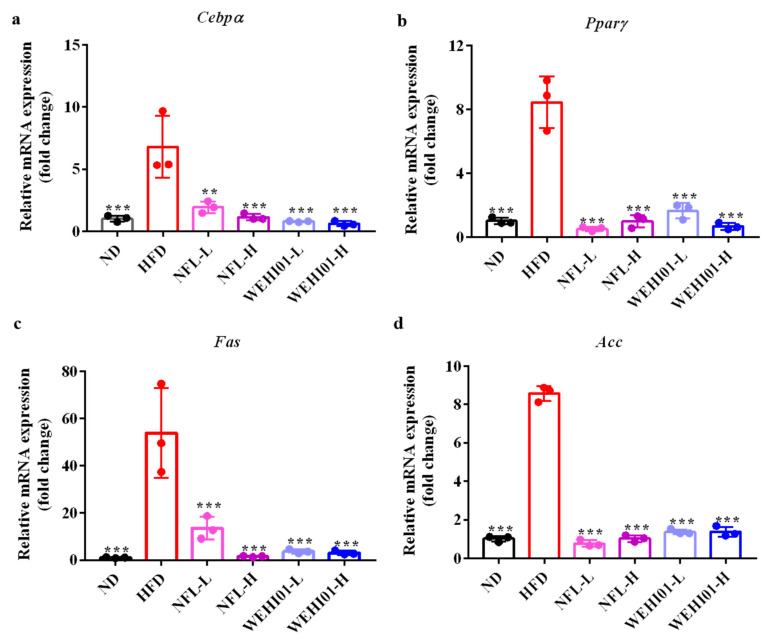
Effect of FLLS-WEHI01 on mRNA expression levels of adipogenic genes in adipose tissue of obese rats (*n* = 3). The mRNA level of (**a**) *Cebpα*, (**b**) *Pparγ*, (**c**) *Fas*, and (**d**) *Acc*. The *β-actin* gene was used as reference gene. Values were expressed as mean ± SD (*n* = 3)., ** *p* < 0.01, and *** *p* < 0.001 vs. HFD group.

**Figure 9 nutrients-14-04348-f009:**
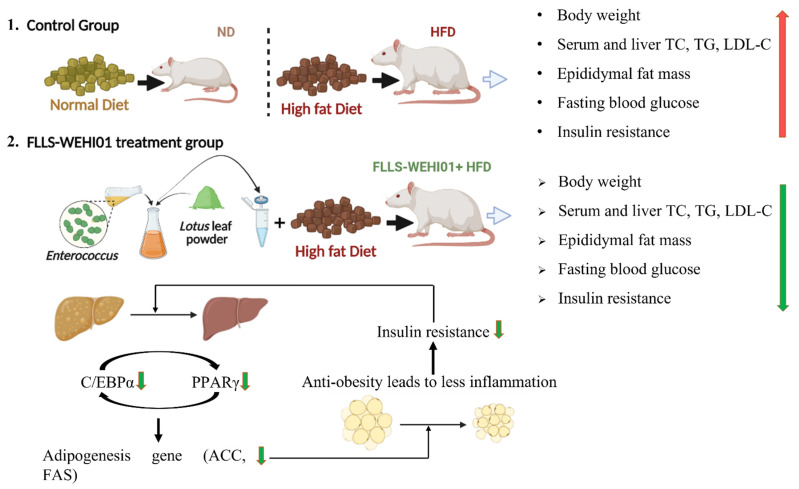
The schematic diagram of the anti-obesity ability of WEHI01-fermented lotus leaf supernatant. ND, normal diet; HFD, high-fat diet; FLLS-WEHI01, *Enterococcus hirae* WEHI01-fermented lotus leaf supernatant; TC, total cholesterol; TG, total triglyceride; LDL-C, low-density lipoprotein cholesterol; C/EBPα, CCAAT/enhancer-binding protein α; PPARγ, peroxisome proliferator activated receptors γ; ACC, acetyl-CoA carboxylase; FAS, fatty acid synthase.

**Table 1 nutrients-14-04348-t001:** Effect of FLLS-WEHI01 on liver, epididymis, kidney, and spleen fat weight as well as food intake in HFD-induced obese mice.

Weight (g)	ND	HFD	NFL-L	NFL-H	WEHI01-L	WEHI01-H
Liver	11.13 ± 0.93 ***	20.37 ± 3.10	20.31 ± 1.63	20.42 ± 2.50	18.08 ± 1.26 *	16.78 ± 0.77 *
Epididymis fat	2.15 ± 0.68 *	4.69 ± 1.15	3.83 ± 1.45	3.72 ± 1.22	3.30 ± 0.96	2.84 ± 0.64 *
Kidney	2.11 ± 0.19	2.31 ± 0.22	2.31 ± 0.11	2.24 ± 0.16	2.09 ± 0.18	2.10 ± 0.21
Spleen	0.59 ± 0.09	0.89 ± 0.20	1.00 ± 0.31	1.01 ± 0.19	0.87 ± 0.11	0.95 ± 0.22
Food intake	20.99 ± 1.48	19.90 ± 2.37	20.06 ± 2.76	19.87 ± 3.22	19.06 ± 3.42	19.80 ± 2.39

Values presented are the mean ± SD (*n* = 7/group). * *p* < 0.05 and *** *p* < 0.001 vs. HFD.

## Data Availability

The data that support the findings of this study are available from the corresponding author, [author initials], upon reasonable request.

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
