# Peer review of "Lotus (Nelumbo nucifera Gaertn.) Leaf-Fermentation Supernatant Inhibits Adipogenesis in 3T3-L1 Preadipocytes and Suppresses Obesity in High-Fat Diet-Induced Obese Rats"

_nutrients, 2022, doi:10.3390/nu14204348_

Round 1
Reviewer 1 Report
Yao He et al. have evaluated the effect of Lotus (Nelumbo nucifera Gaertn) fermentation on adipogenesis and anti-obesity capacity. For this they have carried out a fermentation with two types of probiotics (Enterococcus faecium WEFA23 and Enterococcus hirae WEH101). They have observed that this extract reduces the capacity of mouse pre-adipocytes to capture lipids and has repressed the expression of some markers of adipogenesis. Furthermore, in an in vivo male rat model of obesity, treatment with the fermented extract (WEH101) has reduced body weight, liver weight, epididymal adipose tissue, improved lipid profile, and improved glucose tolerance. via OGTT. The study is interesting, well planned and with a certain translational capacity that demonstrates the beneficial effect of the fermentation of a plant already used in traditional Eastern Asian medicine. However, the study presents some doubts that must be resolved by the authors.
Minor comments
- Line 150 appears a double space
- Indicate in the statistical analysis section when one-way ANOVA or two-way ANOVA is used.
- In the legend of figure 2, indicate that (a) are the representative images and (b) is the quantification by spectrophotometer.
- The nomenclature of rat genes is not written in capital letters and in italics, but the first in capital letters and the rest in lowercase, all in italics. Change it throughout the manuscript.
- In the legend of figure 5 (line 291). There is an error in the word insulin (appears as insu-lin).
- Line 372 the Verón HE reference, there is an error.
- In the legend of figure 4 the significance of **p<0.01 is indicated, however it does not appear in the figures.
Major comments
- Why have the authors only used males for the in vivo experiments?
- Why NFLLS and WEH101 was only given in animals treated with HFD and not in animals fed with a standard diet, to see the effect of the extracts by itself.
- The epididymal adipose tissue was analyzed by histology, why was the subcutaneous adipose tissue not analyzed? Metabolically, the subcutaneous tissue is completely different from visceral tissue such as the epididymal.
- In the RT-PCR, beta actin was used as housekeeping, however different authors report a variability of beta actin as housekeeping in in vivo models of obesity (PMID: 28620170). Is there variability of this gene in the PCR analysis?
- In lines 202-203 the authors comment “Results were expressed as the mean ± standard deviation (SD) of three independent experiments.” What does it mean? Three replicates? Or that they have repeated it 3 times with different replicates? The authors should be more precise indicating the n of each experiment. Furthermore, in the case of in vivo animals, the n per group is 8 individuals, so this is not indicated in the statistical analysis section.
- The authors show the figures as bar graphs, representing the mean +/- SD. However, it would help a lot to see the dispersion of the data and the number of each replicates, if the individual data are shown representing the graphs with “Scatter plot with bar” in all the graphs.
- Figure 1 shows the effect of the extracts on cell viability. I have some questions about this analysis. The first of them is that the control on which 100% of the viability is made is a group with treatment of the non-fermented extracts at low doses, instead of a group in which no treatment has been added (only the vehicle) which is what would correspond to a real control group with which to make comparisons. The second issue is that all the groups seem to have more viability than the NFL-L group, particularly the NFL-H and VEH101-H group. Is this increase in viability statistically significant? If it is significant, what could be the cause? Is it increasing cell proliferation? A cell cycle analysis would be interesting if there is an increase in proliferation.
- Figure 2 shows the effect of the extracts on the ability of pre-adipocytes to accumulate lipids. The differences observed in panel 2b do not correspond with the representative images of panel 2a, for example, the NFL-H group is lower than the NFL-L group, however in panel 2a, it seems that the NFL-L has less oil-red staining than NFL-H. The representative images should correspond to what was observed in the quantification of panel 2b. Another issue is the statistical analysis, the NFL-H group is lower than the control group (around 80%, indicate in the text the percentages and p of these groups as indicated in the rest of the groups on lines 223- 224). However, no significant difference is observed in the graph, is this correct? From what is observed with the mean and SD, it seems that it could be statistically significant.
- Figure 3 presents several questions. First, I have doubts if there are all the significant differences indicated in the figure, for example the expression of Cebpa (C/EBPa) is repressed in pre-adipocytes treated with NFL-H, however no significant difference is indicated. Regarding the expression of Cepbb, NFL-L and NFL-H have the same levels of expression as WEH101-L, however, they do not show differences compared to the control. Check and indicate all significant differences in all graphs that appear in the manuscript, as well as the p of each experimental group. This possible error is observed in almost all the graphs.
- It is recommended to normalize gene expression with more than one housekeeping. The analysis with only one compromised the results.
- The authors have evaluated the expression of 4 adipongenesis markers (Cebpa, Cebpb, Fas, Acc), however there are other markers that can provide more information, such as Cepbd, Adipoq, Lpl, Lep, Pparg and Aqp7. All of them can provide information if these extracts reduce adipogenesis and if this reduction is specific to one of the stages of the adipogenesis process.
- Figure 4 shows the effect of the extracts (fermented and non-fermented) in an in vivo model. It is observed how all the treatments reduce body weight, and only the animals treated with WEH101-H also reduce the weight of the liver and epididymal adipose tissue. Did the animals eat the same amount of food between groups? It would be very helpful to show how these animals ate during the treatment, since these effects may simply be due to the animals eating less. Showing the weight of the rest of the adipose tissues would be highly recommended. It would also be helpful to show the weight of other tissues such as heart, muscle, and other types of adipose tissue to see the general status and see if there may be an accumulation of lipids in other tissues.
- In figure 5a, the authors show the effect of the different treatments after the OGTT. Significance ND vs HFD and WEH101-L vs HFS are shown. Do these differences correspond to the last measurement point? or this difference correspond to all time points? It would be very helpful to analyze the AUC of all the groups, it is possible that other differences will be observed.
- Figure 5b shows the fasting glucose levels, with the levels of the HFD group close to 13mM, however, in Figure 5a this same group, at point 0, have levels close to the rest of the group (6mM). Explain this difference.
- TG levels are reduced in all groups compared to the HFD group, however only the WEH101-H group has reduced liver and adipose tissue weight. Why is it that fasting TG levels are reduced in all groups and tissues do not change? Is it activating beta activation in other tissues? It would be relevant to see if the beta oxidation of fatty acids is activated in the liver and muscle. On the other hand, have plasmatic NEFA levels been measured?
- Figure 7b, the NFL-H group seems to have fewer lipids than the HFD group, however this is not seen in panels 6e and 7f. What can this difference be due to?
- Why have adipogenesis genes not been measured in the adipose tissue itself, but in the liver?
- In line 379, the authors say that the treatment with the fermented extract FLLS-WEH101 high concentration decreases the hypertrophy of the epididymal adipocytes, however, their area has not been measured and quantified, but only a representative photo by group. The authors should measure the area of the adipocytes and make a quantitative analysis of it to confirm this fact.
- The authors should confirm if the livers present less accumulation of lipids by quantifying the red oil staining images. They must show a graph with these results.
- On line 409-413. The authors comment “Our results of in vitro analysis showed that FLLS could significantly down-regulate mRNA levels of C/EBPα, C/EBPβ and ACC, FAS in 3T3-L1 cells, which was consistent with previous report that panicin could significantly down-regulate mRNA levels. regulate the mRNA levels of C/EBPα, ACC and FAS, thereby inhibiting 3T3-L1 preadipocyte differentiation and adipogenesis”. However, not all fermented extracts decrease the expression of these markers, moreover, there is an extract NOT fermented (WEFA23-L) that does lower all of them. What is the advantage of the fermented extract vs. the non-fermented extract in the adipogenesis of pre-adipocytes?
- In lines 430-431, the authors say “FLLS-WEHI01 exhibited better inhibitory effect on adipogenesis and lipid accumulation”. By what criteria do you reach that conclusion? If you refer to the results shown in Figure 2b, the fermented extract WEFA23-H shows a similar (or even higher) reduction than WEH101-H. If you refer to the adipogenesis markers shown in figure 3, WEFA23-H is the only extract capable of reducing all the markers and not WEH101-H. Why have the authors chosen the WEH101 to perform the in vivo experiments? Moreover, according to these results, a greater effect can be expected with the WEFA23-H extract. This point is crucial.
Author Response
Dear Editors and Reviewers:
Thanks for your letter and for the reviewers’ comments concerning our manuscript, entitled “Lotus (Nelumbo nucifera Gaertn.) Leaf Fermentation Supernatant Inhibits Adipogenesis in 3T3-L1 Preadipocytes and Suppresses Obesity in High-Fat Diet-Induced Obese Rats” (ID: nutrients-1911079). Those comments are all valuable and very helpful for revising and improving our paper, as well as the important guiding significance to our research. We have comprehensively considered your opinions and tried our best to revise the manuscript according to the Reviewers’ pertinent comments. Revised portions are marked in yellow in the renew manuscript. We hope that the revision will meet with your requirements and looking forward to your positive decision.
Yours sincerely
Dr. Xueying Tao
State Key Laboratory of Food Science and Technology, Nanchang University, Nanchang 330047, P. R. China.
Tel: +86-791-88334578;
Fax: +86-791-8833-3708;
E-mail: [email protected]
The following is a point-to-point response to the reviewers’ comments:
Author’s response to reviewer’s comments
Reviewer #1: General comments
Yao He et al. have evaluated the effect of Lotus (Nelumbo nucifera Gaertn) fermentation on adipogenesis and anti-obesity capacity. For this they have carried out a fermentation with two types of probiotics (Enterococcus faecium WEFA23 and Enterococcus hirae WEH101). They have observed that this extract reduces the capacity of mouse pre-adipocytes to capture lipids and has repressed the expression of some markers of adipogenesis. Furthermore, in an in vivo male rat model of obesity, treatment with the fermented extract (WEH101) has reduced body weight, liver weight, epididymal adipose tissue, improved lipid profile, and improved glucose tolerance. via OGTT. The study is interesting, well planned and with a certain translational capacity that demonstrates the beneficial effect of the fermentation of a plant already used in traditional Eastern Asian medicine. However, the study presents some doubts that must be resolved by the authors.
Minor comments
Point 1: Line 150 appears a double space.
Response 1: Thanks for your kind advice. We have deleted the double space (Line 48).
Point 2: Indicate in the statistical analysis section when one-way ANOVA or two-way ANOVA is used.
Response 2: Thanks for your suggestion. We have indicated clearly that one-way analysis of variance and a Tukey’s multiple comparisons test were used for statistical analysis in the paper and p-values less than 0.05 were considered statistically significant (line 242-245).
Point 3: In the legend of figure 2, indicate that (a) are the representative images and (b) is the quantification by spectrophotometer.
Response 3: Thanks for your kind comment. We have indicated in Fig. 2 that (a) are the representative images and (b) is the quantification by spectrophotometer as you suggested (line 673-677).
Point 4: The nomenclature of rat genes is not written in capital letters and in italics, but the first in capital letters and the rest in lowercase, all in italics. Change it throughout the manuscript.
Response 4: Thanks for your kind advice. We have corrected the gene names throughout the manuscript as you suggested.
Point 5: In the legend of figure 5 (line 291). There is an error in the word insulin (appears as insu-lin).
Response 5: Sorry for our mistake. We have changed the misspelled word “insu-lin” into “insulin” (line 693).
Point 6: Line 372 the Verón HE reference, there is an error.
Response 6: Thanks for your kind suggestion. We found the reference was not perfect to be cited at the manuscript and therefore deleted it.
Point 7: In the legend of figure 4 the significance of **p<0.01 is indicated, however it does not appear in the figures.
Response 7: Thanks for your kind suggestion. We have deleted the indicated significance of **p<0.01 in the legend of figure 4 (line 689).
Major comments
Point 1: Why have the authors only used males for the in vivo experiments?
Response 1: Thanks for your kind advice. It’s known that a suitable animal model with rodents should be as similar as possible to the humans in several respects and should allow the study of biological phenomena, animal behavior, induced pathological process or spontaneous, and thus to provide the understanding of diseases in humans. Fernandes reported that after analyzing and comparing several articles, male rodents were considered being the species most commonly used in experimental studies to induce obesity which were searched by using the following keywords “obesity animal models, diet and rodents” published between the years 2010 to 2016 (Fernandes et al., 2016; Doulberis et al., 2020). In addition, we found male Sprague-Dawley (SD) rat was chosen to establish the obesity model in many articles published in Nutrients with the subject of effect of lactic acid bacteria or lotus in anti-obesity (Kim et al., 2020; Lee et al., 2017). Combined the above two points, to establish a reliable obesity model, male SD rats were selected for the in vivo experiment in our study.
References
Fernandes, M.R.; Lima, N.V.d.; Rezende, K.S.; Santos, I.C.M.; Silva, I.S.; Guimarães, R.d.C.A. Animal models of obesity in rodents. An integrative review1. Acta cirurgica brasileira 2016, 31, 840-844.
Doulberis, M.; Papaefthymiou, A.; Polyzos, S.A.; Katsinelos, P.; Grigoriadis, N.; Srivastava, D.S.; Kountouras, J. Rodent models of obesity. Minerva endocrinologica 2020, 45, 243-263.
Kim, H.Y.; Hong, M.H.; Kim, K.W.; Yoon, J.J.; Lee, J.E.; Kang, D.G.; Lee, H.S. Improvement of hypertriglyceridemia by roasted Nelumbinis folium in high fat/high cholesterol diet rat model. Nutrients 2020, 12, 3859.
Lee, S.G.; Lee, Y.J.; Jang, M.-H.; Kwon, T.R.; Nam, J.-O. Panax ginseng leaf extracts exert anti-obesity effects in high-fat diet-induced obese rats. Nutrients 2017, 9, 999.
Point 2: Why NFLLS and WEHI01 was only given in animals treated with HFD and not in animals fed with a standard diet, to see the effect of the extracts by itself.
Response 2: Thanks for your valuable suggestion. The purpose of our study was to investigate whether the fermentation of lotus by E. hirae WEHI01 could enhance its anti-obesity ability. Therefore, we used the unfermented lotus (NFLLS) and the lotus fermented with WEHI01 (FLLS-WEHI01) to compare their effect of anti-obesity in HFD-induced obesity model. As for treatments with NFLLS and FLLS-WEHI01 in animals fed with a standard diet, we do think this is a good topic to research, but perhaps a bit unrelated to the anti-obesity topic in this study.
Point 3: The why was the subcutaneous adipose tissue not analyzed? Metabolically, the subcutaneous tissue is completely different from visceral tissue such as the epididymal.
Response 3: Thanks for your kind advice. In fact, epididymal adipose tissue belongs to visceral adipose tissue (VAT), which is more cellular, vascular, innervated and contains a larger number of inflammatory and immune cells, lesser preadipocyte differentiating capacity and a greater percentage of large adipocytes compared with subcutaneous adipose tissue (SCAT). It’s reported that there contained more glucocorticoid and androgen receptors in VAT compared with SCAT, which indicated that VAT adipocytes are more metabolically active, more sensitive to lipolysis and more insulin-resistant than SCAT adipocytes (Ibrahim et al., 2010). Therefore, we analyzed epididymal adipose tissue by histology in our research.
References
Ibrahim, M.M. Subcutaneous and visceral adipose tissue: structural and functional differences. Obesity reviews 2010, 11, 11-18.
Point 4: In the RT-PCR, beta actin was used as housekeeping, however different authors report a variability of beta actin as housekeeping in in vivo models of obesity (PMID: 28620170). Is there variability of this gene in the PCR analysis?
Response 4: Thanks for your valuable suggestion. We have read the corresponding reference you referred in the review, beta actin is indeed reported to be variable in animal models. We checked all of our initial data of RT-qPCR and found no variability in the data (the Ct values of housekeeping gene beta actin were all about 18). We will use multiple housekeeping genes to perform the RT-qPCR to make the results more objective in the future experiment.
Point 5: In lines 202-203 the authors comment “Results were expressed as the mean ± standard deviation (SD) of three independent experiments.” What does it mean? Three replicates? Or that they have repeated it 3 times with different replicates? The authors should be more precise indicating the n of each experiment. Furthermore, in the case of in vivo animals, the n per group is 8 individuals, so this is not indicated in the statistical analysis section.
Response 5: Thanks for your kind comment. “Results were expressed as the mean ± standard deviation (SD) of three independent experiments” means that we performed the experiments 3 times with different replicates. In addition, we have indicated the n value of each experiment precisely as you suggested.
Point 6: The authors show the figures as bar graphs, representing the mean +/- SD. However, it would help a lot to see the dispersion of the data and the number of each replicates, if the individual data are shown representing the graphs with “Scatter plot with bar” in all the graphs.
Response 6: Thank for your valuable suggestion. We have corrected the “bar graphs” into “Scatter plot with bar” in all the graphs to better see the dispersion of the data and the number of each replicate.
Point 7: Figure 1 shows the effect of the extracts on cell viability. I have some questions about this analysis. The first of them is that the control on which 100% of the viability is made is a group with treatment of the non-fermented extracts at low doses, instead of a group in which no treatment has been added (only the vehicle) which is what would correspond to a real control group with which to make comparisons. The second issue is that all the groups seem to have more viability than the NFL-L group, particularly the NFL-H and VEH101-H group. Is this increase in viability statistically significant? If it is significant, what could be the cause? Is it increasing cell proliferation? A cell cycle analysis would be interesting if there is an increase in proliferation.
Response 7: Thanks for your valuable suggestion. In fact, the control of which 100% of the viability was set by a group with only the vehicle and no treatment been added (the data was not shown as a bar in the Figure 1). As for the groups with treatment of the non-fermented extracts at low doses, it was not set as the control and was actually an experimental group with the viability less than 100% (97.8%). In addition, all the experimental groups showed some differences compared with the control group (the viability of the group is 100%) but indicated no statistical significance, and this is why all the groups were chosen to investigate for the following experiments.
Point 8: Figure 2 shows the effect of the extracts on the ability of pre-adipocytes to accumulate lipids. The differences observed in panel 2b do not correspond with the representative images of panel 2a, for example, the NFL-H group is lower than the NFL-L group, however in panel 2a, it seems that the NFL-L has less oil-red staining than NFL-H. The representative images should correspond to what was observed in the quantification of panel 2b. Another issue is the statistical analysis, the NFL-H group is lower than the control group (around 80%, indicate in the text the percentages and p of these groups as indicated in the rest of the groups on lines 223- 224). However, no significant difference is observed in the graph, is this correct? From what is observed with the mean and SD, it seems that it could be statistically significant.
Response 8: Thanks for your kind suggestion. The mouse 3T3-L1 preadipocytes were cultured and induced to differentiate by adipocyte differentiation cocktail and insulin until lipid droplet formation was observed, following by fixing the cells with 4% polyformaldehyde for 30 min, and finally the stained lipid droplets were observed by using a light microscope (Olympus, Japan) (Figure 2a), and the droplets were then dissolved in isopropanol and at 530 nm using a spectrophotometer (Figure 2b). Therefore, we can know that what presented in Figure 2a were part of the complete vision and that’s to say the quantified data by measuring the absorbance were not necessarily completely march with part vision presented in Figure 2a. As for the statistical analysis, there did not exist significant difference between the NFL-H and control group (p = 0.08). In addition, we have indicated in the text the percentages and p of these groups where necessary.
Point 9: Figure 3 presents several questions. First, we have doubts if there are all the significant differences indicated in the figure, for example the expression of Cebpa (C/EBPa) is repressed in pre-adipocytes treated with NFL-H, however no significant difference is indicated. Regarding the expression of Cebpb, NFL-L and NFL-H have the same levels of expression as WEH101-L, however, they do not show differences compared to the control. Check and indicate all significant differences in all graphs that appear in the manuscript, as well as the p of each experimental group. This possible error is observed in almost all the graphs.
Response 9: Thanks for your kind advice. We have checked and indicated all significant differences in all graphs that appear in the manuscript, as well as the p of each experimental group as you suggested. Firstly, we feel very sorry for that we did ignored to indicate the significant difference between NFL-H and control group (p = 0.02) in the expression of Cebpα. Secondly, for the statistical analysis on the expression of Cebpβ, there did not exist any error, in which the p value between NFL-L, NFL-H and control group achieved 0.0684 and 0.0557, respectively.
Point 10: It is recommended to normalize gene expression with more than one housekeeping. The analysis with only one compromised the results.
Response 10: Thanks for your kind advice. It’s reported that a variability of beta actin as housekeeping in in vivo models of obesity (PMID: 28620170). The beta actin was used as housekeeping gene in our experiment, and we have checked our initial data of RT-PCR and found there didn’t exit variability of the data (the ct value of housekeeping gene beta actin was about 18). In addition, we will use more than one house keeping genes to perform the RT-PCR experiment in in vivo models of obesity.
Point 11: The authors have evaluated the expression of 4 adipongenesis markers (Cebpa, Cebpb, Fas, Acc), however there are other markers that can provide more information, such as Cepbd, Adipoq, Lpl, Lep, Pparg and Aqp7. All of them can provide information if these extracts reduce adipogenesis and if this reduction is specific to one of the stages of the adipogenesis process.
Response 11: Thanks for your kind comment. Our aim of the paper was to investigate whether fermentation with Enterococcus could enhance the anti-obesity of lotus leaf, which was still at the phenomenon level. Therefore, we choose to evaluate the mRNA expression of 4 adipongenesis markers (Cebpa, Cebpb, Fas, Acc) to reflect the capacity of anti-obesity. In addition, we are planning to dig out the potential mechanism of the supernatant of Enterococcus fermented lotus leaf next step, therefore mRNA and protein levels of other markers such as Cepbd, Adipoq, Lpl, Lep, Pparg and Aqp7 will be evaluated in the plan to reveal the concrete mechanism.
Point 12: Figure 4 shows the effect of the extracts (fermented and non-fermented) in an in vivo model. It is observed how all the treatments reduce body weight, and only the animals treated with WEHI01-H also reduce the weight of the liver and epididymal adipose tissue. Did the animals eat the same amount of food between groups? It would be very helpful to show how these animals ate during the treatment, since these effects may simply be due to the animals eating less. Showing the weight of the rest of the adipose tissues would be highly recommended. It would also be helpful to show the weight of other tissues such as heart, muscle, and other types of adipose tissue to see the general status and see if there may be an accumulation of lipids in other tissues.
Response 12: Thanks for your kind suggestion. Food intake of animals was showed in Figure S1, we can easily see that there were no significant differences between groups. In addition, WEHI01-H showed the significant decrease (p < 0.001) as early as in the second week and this trend retained till the end of the experiment while other treatments didn’t achieve this effects as we can know from Figure 4a. Maybe the two points combined could explain why only rats treated with WEHI01-H obtained a reduced weight of the liver and epididymal adipose tissue. In addition, we also detected the spleen and kidney weight of rats and the results were showed in Figure S2 and there existed no significant differences between groups.
Figure S1. Effect of FLLS-WEHI01 on food intake of obese rats. Values were expressed as mean ± SD (n = 6).
Figure S2. Effect of supernatant of FLLS-WEHI01 on organ index of obese rats. (a) Kidney index; (b) Spleen index. Values were expressed as mean ± SD (n = 7).
Point 13: In figure 5a, the authors show the effect of the different treatments after the OGTT. Significance ND vs HFD and WEH101-L vs HFS are shown. Do these differences correspond to the last measurement point? or this difference correspond to all time points? It would be very helpful to analyze the AUC of all the groups, it is possible that other differences will be observed.
Response 13: Thanks for your kind comment. The significance ND vs HFD and WEHI01-L vs HFS were correspond to the last measurement point. The AUC of all the groups were analyzed and presented in Figure S3, and there were no significant differences between all the groups.
Figure S3. The area under curve (AUC) of oral glucose tolerance test of obese rats. Values were expressed as mean ± SD (n = 3).
Point 14: Figure 5b shows the fasting glucose levels, with the levels of the HFD group close to 13 mM, however, in Figure 5a this same group, at point 0, have levels close to the rest of the group (6 mM). Explain this difference.
Response 14: Thanks for your kind comment. The serum used to measure glucose levels at point 0 were collected by tail vein while the serum used to determine fasting glucose levels showed in Figure 5b were collected by eyeball blood, this is why there existed some distinction between the two values. Nevertheless, Figure 5a or 5d were reliable enough to reveal the effects and differences between all the groups cause all the samples in Figure 5a or 5b were collected by the same methods.
Point 15: TG levels are reduced in all groups compared to the HFD group, however only the WEH101-H group has reduced liver and adipose tissue weight. Why is it that fasting TG levels are reduced in all groups and tissues do not change? Is it activating beta activation in other tissues? It would be relevant to see if the beta oxidation of fatty acids is activated in the liver and muscle. On the other hand, have plasmatic NEFA levels been measured?
Response 15: Thanks for your suggestion. As mentioned above, treatment with WEHI01-H significantly decreased the body weight of HFD-induced obese rats as early as the second week. This may partly explain why only the WEH101-H group had reduced liver and adipose tissue weight while TG levels were reduced in all groups compared to the HFD group. Similarly, Jin demonstrated that all doses of polymethoxyflavone-rich fraction (PRF) from Jinkyool (Citrus sunki Hort. ex Tanaka) Leaf were capable to decrease the TG content in rat’s serum while not decrease the liver weight of all groups (Jin et al., 2022). On the other hand, we should further determine plasmatic NEFA levels and investigate whether the beta oxidation of fatty acids is activated in the liver and muscle, which will better explain our results.
Reference
Jin, Y.-J.; Jang, M.-G.; Kim, J.-W.; Baek, S.; Ko, H.-C.; Hur, S.-P.; Kim, S.-J. Anti-Obesity Effects of Polymethoxyflavone-Rich Fraction from Jinkyool (Citrus sunki Hort. ex Tanaka) Leaf on Obese Mice Induced by High-Fat Diet. Nutrients 2022, 14, 865.
Point 16: Figure 7b, the NFL-H group seems to have fewer lipids than the HFD group, however this is not seen in panels 6e and 7f. What can this difference be due to?
Response 16: Thanks for your kind suggestion. As we can observe in Figure 7b, the NFL-H group seems to have fewer lipids than the HFD group, which was consistent with the result in Figure 6e (treatment with NFL-H showed a significant decrease compared with HFD (p = 0.04)). Contrary, the result in Figure 6f seems a little discrepancy with what presented in Figure 7b, we speculated it was because the vision presented was only one only one part of the whole vision in the NFL-H group, which could partly reveal the effect of treatments.
Point 17: Why have adipogenesis genes not been in the adipose tissue itself, but in the liver?
Response 17: Thanks for your kind comment. We have measured the mRNA expression of adipogenesis genes both in liver (the data were not showed in manuscript) and the epididymal adipose tissue of obese rats and the data presented in the manuscript were indeed the mRNA level of adipogenesis genes in the epididymal adipose tissue, we feel very sorry for confusing the information and we have corrected the relative information.
Point 18: In line 379, the authors say that the treatment with the fermented extract FLLS-WEH101 high concentration decreases the hypertrophy of the epididymal adipocytes, however, their area has not been measured and quantified, but only a representative photo by group. The authors should measure the area of the adipocytes and make a quantitative analysis of it to confirm this fact.
Response 18: Thanks for your kind suggestion. We made a quantitative analysis of the size of adipose cells according to Kho’s report (Kho et al., 2016). As we can observe in Figure S4, HFD treatment significant increased the size of adipose (p = 0.0081) while treatment with NFL-H, WEHI01-L, and WEHI01 could significant decrease the size of adipose cells (p = 0.0392, 0.0282, and 0.0220 respectively).
Figure S4. The effect of FLLS-WEHI01 on adipose cells size of obese rats. Values were expressed as mean ± SD (n = 3). *p < 0.05, **p < 0.01 and ***p < 0.001 vs. HFD group.
Reference
Kho, M.C.; Lee, Y.J.; Park, J.H.; Kim, H.Y.; Yoon, J.J.; Ahn, Y.M.; Tan, R.; Park, M.C.; Cha, J.D.; Choi, K.M. Fermented red ginseng potentiates improvement of metabolic dysfunction in metabolic syndrome rat models. Nutrients 2016, 8, 369.
Point 19: The authors should confirm if the livers present less accumulation of lipids by quantifying the red oil staining images. They must show a graph with these results.
Response 19: Thanks for your kind advice. We made a quantitative analysis of the size of adipose cells according to Kho’s report [1]. As we can observe in Figure S5, HFD treatment significant increased the size of adipose (p = 0.0012) while treatment with NFL-H, WEHI01-L, and WEHI01 could significant decrease the size of adipose cells (p = 0.0431, and 0.0144 respectively).
Figure S6. The effect of FLLS-WEHI01 on fat accumulation of obese rats. Values were expressed as mean ± SD (n = 3). *p < 0.05, **p < 0.01 and ***p < 0.001 vs. HFD group.
Point 20: On line 409-413. The authors comment “Our results of in vitro analysis showed that FLLS could significantly down-regulate mRNA levels of C/EBPα, C/EBPβ and ACC, FAS in 3T3-L1 cells, which was consistent with previous report that panicin could significantly down-regulate mRNA levels. regulate the mRNA levels of C/EBPα, ACC and FAS, thereby inhibiting 3T3-L1 preadipocyte differentiation and adipogenesis”. However, not all fermented extracts decrease the expression of these markers, moreover, there is an extract NOT fermented (WEFA23-L) that does lower all of them. What is the advantage of the fermented extract vs. the non-fermented extract in the adipogenesis of pre-adipocytes?
Response 20: Thanks for your kind suggestion. Actually, the research on lowering lipid of lotus leaf fermentation liquid is rare, and the function of lowering lipid of lotus leaf were mainly focused on active ingredients, including polysaccharides, flavonoids, alkaloids, polyphenols, etc [2]. For example, many scholars reported that lotus leaf alkaloids can control weight, prevent obesity, regulate lipid metabolism disorder, inhibit pancreatic lipase activity, and have a time and concentration dependent inhibition effect in the 3T3-L1 preadipocyte model in vitro (Wang et al., 2021; Park et al., 2022). Besides, polysaccharide was reported to possess the capacity to inhibit obesity from many aspects such as modulation of intestinal flora or regulation of inflammation (Sang et al., 2021; Wu et al., 2019). Indeed, we also compared the differences of the bioactive substances content in lotus before and after fermentation with Enterococcus by HPLC. As we can observe in Figure S6 showed below, fermentation with WEHI01 and WEFA23 could significantly increase the crude polysaccharide content of the lotus leaf supernatant, and fermentation with WEHI01 could also increase the content of short chain fatty acids in the lotus leaf supernatant. What we mentioned above was the advantage of the fermented extract vs. the non-fermented extract in the adipogenesis of pre-adipocytes.
Figure S6. The effect of fermentation with WEHI01 and WEFA23 on content of crude polysaccharide and SCFAs in lotus leaf supernatant. (a) Content of crude polysaccharide; (b) content of acetic acid; (c) content of propionic acid; (d) content of butyric acid. Values were expressed as mean ± SD (n = 3). *p < 0.05, **p < 0.01 and ***p < 0.001 vs. NFL-L group.
#p < 0.05, ##p < 0.01 and ###p < 0.001 vs. NFL-H group.
Zheng, H.; Han, L.; Shi, W.; Fang, X.; Hong, Y.; Cao, Y. Research advances in lotus leaf as chinese dietary herbal medicine. The American Journal of Chinese Medicine 2022, 50, 1423-1445.
Wang, Z.; Li, Y.; Ma, D.; Zeng, M.; Wang, Z.; Qin, F.; Chen, J.; Christian, M.; He, Z. Alkaloids from lotus (Nelumbo nucifera): Recent advances in biosynthesis, pharmacokinetics, bioactivity, safety, and industrial applications. Critical Reviews in Food Science and Nutrition 2021, 1-34.
Park, M.; Han, J.; Lee, H.-J. Anti-adipogenic effect of neferine in 3T3-L1 cells and primary white adipocytes. Nutrients 2020, 12, 1858.
Sang, T.; Guo, C.; Guo, D.; Wu, J.; Wang, Y.; Wang, Y.; Chen, J.; Chen, C.; Wu, K.; Na, K. Suppression of obesity and inflammation by polysaccharide from sporoderm-broken spore of Ganoderma lucidum via gut microbiota regulation. Carbohydrate polymers 2021, 256, 117594.
Wu, T.-R.; Lin, C.-S.; Chang, C.-J.; Lin, T.-L.; Martel, J.; Ko, Y.-F.; Ojcius, D.M.; Lu, C.-C.; Young, J.D.; Lai, H.-C. Gut commensal Parabacteroides goldsteinii plays a predominant role in the anti-obesity effects of polysaccharides isolated from Hirsutella sinensis. Gut 2019, 68, 248-262.
Point 21: In lines 430-431, the authors say “FLLS-WEHI01 exhibited better inhibitory effect on adipogenesis and lipid accumulation”. By what criteria do you reach that conclusion? If you refer to the results shown in Figure 2b, the fermented extract WEFA23-H shows a similar (or even higher) reduction than WEH101-H. If you refer to the adipogenesis markers shown in figure 3, WEFA23-H is the only extract capable of reducing all the markers and not WEH101-H. Why have the authors chosen the WEH101 to perform the in vivo experiments? Moreover, according to these results, a greater effect can be expected with the WEFA23-H extract. This point is crucial.
Response 21: Thanks for your kind comment. The reason why we choose WEHI01 as the starter to ferment lotus could be induced into three points. Firstly, WEHI01-H achieved better inhibitory effect on adipogenesis and lipid accumulation (inhibition rate up to 56.17%) in 3T3-L1 cells, and it can achieve a significant effect even at a lower dose (p=0.008), which performed better than WEFA23 did. Similar results were also occurred in mRNA level of adipogenesis markers shown in figure 3. Secondly, our previously study showed that E. hirae WEHI01 was capacity to relieve type 2 diabetes (Wei et al., 2020), which was often accompanied with the occurrence and development of obesity. Finally, as we showed in Figure S6, fermentation with WEHI01 could improve the SCFAs content of lotus leaf supernatant, which were thought possessed the ability to anti-obesity. Combined this, we finally choose the FLLS-WEHI01 to perform the in vivo experiments.
Wei, M.; Gu, E.; Luo, J.; Zhang, Z.; Xu, D.; Tao, X.; Shah, N.P.; Wei, H. Enterococcus hirae WEHI01 isolated from a healthy Chinese infant ameliorates the symptoms of type 2 diabetes by elevating the abundance of Lactobacillales in rats. Journal of dairy science 2020, 103, 2969-2981.
Reviewer 2 Report
The manuscript entitled „Lotus (Nelumbo nucifera Gaertn.) Leaf Fermentation Supernatant Inhibits Adipogenesis in 3T3-L1 Preadipocytes and Suppresses Obesity in High-Fat Diet-Induced Obese Rats” presents interesting issue, but some problems should be corrected.
General:
The English language should be polished – it seems as Authors are not native English speakers and some sentences are hard to follow and to understand – preferable they should have their manuscript polished by a professional English correcting agency.
Abstract:
The aim of the study should be presented (e.g. “The aim was…”)
Authors should clearly describe the studied sample of rats.
Authors should present specific numeric results accompanied by the results of the statistical analysis (p-Values).
Keywords:
Authors should present some additional keywords - other than only words used in their title.
Introduction:
“Multiple therapeutic methods with which to treat obesity are available, including lifestyle intervention, formula diet, drugs as well as bariatric surgery” – Authors should define “formula diet” and should present the approaches listed as those more recommended (lifestyle and dietary interventions) and those to be applied if the other are not effective only (bariatric surgery, pharmacotherapy)
Line 44 – negative side effects should be described
The functional foods other than lotus should be at least listed and their action should be described
Specific mechanisms of lotus actions should be described
Lines 65-95 – this paragraph is messy and it presents some random studies not associated with each other – Authors should present only necessary information
Instead of what was done (“we performed…”), the aim of the study should be presented (e.g. “The aim was…”).
Materials and methods:
It seems that Authors prepared only 1 batch of lotus (without any repetitions, or samples from various batches) – it should be clearly presented
The source of lotus should be described
Authors should indicate if animals were housed together or separately and how it could have influenced experienced stress
It seems that Authors did not verify normality of distribution.
Authors should verify normality of distribution and only for parametric data they should present mean and SD, while for non-parametric they should present median, min and max values.
Authors should use statistical tests based on the distributions observed.
Results:
It seems that Authors did not verify normality of distribution.
Authors should verify normality of distribution and only for parametric data they should present mean and SD, while for non-parametric they should present median, min and max values.
Authors should use statistical tests based on the distributions observed.
Instead of figures (charts), Authors should rather present tables, as the results are hard to follow
Discussion:
In this section, Authors presented a number of information which do not discuss the results of the study in any way (e.g. lines 347-353, lines 358-364, etc). Instead, Authors should focus on specific results.
Authors should in their discussion include 3 areas: (1) compare gathered data with the results by other authors, (2) formulate implications of the results of their study and studies by other authors, (3) formulate the future areas which should be studied.
Limitations of the study should be presented.
It seems that Authors prepared only 1 batch of lotus (without any repetitions, or samples from various batches) – it should be presented as a limitation of the study.
Figure 9 should not be presented in the Discussion Section.
Conclusions:
Authors should focus on the conclusions formulated directly based on their study.
Authors’ contributions:
It seems that contribution of YT, LQ, WX, XH, HW was only minor and they did not participate in preparing manuscript. There is a serious risk of a guest authorship procedure which is forbidden. In such case they should be rather presented in Acknowledgements Section and not be indicated as authors of the study.
Author Response
Dear Editors and Reviewers:
Thanks for your letter and for the reviewers’ comments concerning our manuscript, entitled “Lotus (Nelumbo nucifera Gaertn.) Leaf Fermentation Supernatant Inhibits Adipogenesis in 3T3-L1 Preadipocytes and Suppresses Obesity in High-Fat Diet-Induced Obese Rats” (ID: nutrients-1911079). Those comments are all valuable and very helpful for revising and improving our paper, as well as the important guiding significance to our research. We have comprehensively considered your opinions and tried our best to revise the manuscript according to the Reviewers’ pertinent comments. Revised portions are marked in yellow in the renew manuscript. We hope that the revision will meet with your requirements and looking forward to your positive decision.
Yours sincerely
Dr. Xueying Tao
State Key Laboratory of Food Science and Technology, Nanchang University, Nanchang 330047, P. R. China.
Tel: +86-791-88334578;
Fax: +86-791-8833-3708;
E-mail: [email protected]
The following is a point-to-point response to the reviewers’ comments:
Author’s response to reviewer’s comments
Reviewer #2: General comments
The manuscript entitled “Lotus (Nelumbo nucifera Gaertn.) Leaf Fermentation Supernatant Inhibits Adipogenesis in 3T3-L1 Preadipocytes and Suppresses Obesity in High-Fat Diet-Induced Obese Rats” presents interesting issue, but some problems should be corrected.
General:
Point 1: The English language should be polished – it seems as Authors are not native English speakers and some sentences are hard to follow and to understand – preferable they should have their manuscript polished by a professional English correcting agency.
Response 1: Thanks for your valuable suggestion. We have carefully checked and improved the English writing in the revised manuscript.
Abstract:
Point 1: The aim of the study should be presented (e.g. “The aim was…”)
Response 1: Thanks for your kind suggestion. We have presented our aim in the abstract with the sentence “In present study, lotus leaves were fermented by two probiotics, Enterococcus faecium WEFA23 and Enterococcus hirae WEHI01, and the anti-adipogenic effect of Enterococcus fermented lotus leaf supernatant (FLLS) were evaluated in 3T3-L1 preadipocytes with the aim to explore whether the anti-obesity capacity of it will enhance after fermentation with Enterococcus and dig out the potential and corresponding mechanism” (line 23-28).
Point 2: Authors should clearly describe the studied sample of rats.
Response 2: Thanks for your kind suggestion. We have added the clear descriptions about the studied sample of rats as you suggested (line 31).
Point 3: Authors should present specific numeric results accompanied by the results of the statistical analysis (p-Values).
Response 3: Thanks for your kind comment. We have presented specific numeric results accompanied by the results of the statistical analysis (p-Values) in the text of the manuscript and the revised specific numeric results were highlighted with color yellow which mainly centered at results section (such as line 261-262 and 275-278).
Keywords:
Point 1: Authors should present some additional keywords - other than only words used in their title.
Response 1: Thanks for your kind comment. We have added word that do not appear in the title, and the updated keywords are as following: lotus leaf; Enterococcus; fermentation; adipogenesis; anti-obesity; dietary supplement; functional foods (line 44-45).
Introduction:
Point 1: “Multiple therapeutic methods with which to treat obesity are available, including lifestyle intervention, formula diet, drugs as well as bariatric surgery” – Authors should define “formula diet” and should present the approaches listed as those more recommended (lifestyle and dietary interventions) and those to be applied if the other are not effective only (bariatric surgery, pharmacotherapy).
Response 1: Thanks for your thoughtful comment. We have added the supplemental information as you suggested and now the sentence turned into “Multiple therapeutic methods with which to treat obesity are available, including lifestyle intervention, formula diet and drugs as well as bariatric surgery, among which the former two means are mainly for obese people with mild symptoms (body mass index; BMI < 25) while the latter two are aimed at patients with severe obesity (BMI > 32) and other complications such as type 2 diabetes. For all treatment approaches, behavior change techniques, reduction in energy intake, and increasing energy expenditure are required” (line 63-67).
Point 2: Line 44 – negative side effects should be described
Response 2: Thanks for your kind comment. We have added the description about the negative side effects as you advised and now the sentence turned into “However, these therapies often accompany with negative side effects such as poor effect induced repeated obesity as well as the discomfort to other organs such as spleen, kidney and stomach” (line 67-70).
Point 3: The functional foods other than lotus should be at least listed and their action should be described
Response 3: Thanks for your kind advice. We have added the sentence “For example, the oral administration of the fermented garlic extract by Lactobacillus plantarum BL2 (LAFGE) reduced the body weight and epididymal, retroperitoneal, as well as mesenteric adipose tissue mass of high-fat diet male C57BL/6J mice, which was account of LAFGE could inhibit lipogenesis by down-regulating the mRNA and protein expression of Pparγ, Cebpα, and lipogenic proteins” (line 73-78) to present the description about functional foods and their action besides lotus.
Point 4: Specific mechanisms of lotus actions should be described
Response 4: Thanks for your kind suggestion. We have added the specific mechanisms currently reported of lotus caffeine action of anti-obesity in the manuscript and the sentence is “For example, Wang et. al reported that the anti-obesity effects of nuciferine (NUC) may be related to modulations in the composition and potential function of gut microbiota, improvement in intestinal barrier integrity and prevention of chronic low-grade inflammation” (line 93-96).
Point 5: Lines 65-95 – this paragraph is messy and it presents some random studies not associated with each other – Authors should present only necessary information
Response 5: Thanks for your thoughtful comment. We have deleted the messy and non-associated information and only presented the necessary information (line 100-110).
Point 6: Instead of what was done (“we performed…”), the aim of the study should be presented (e.g. “The aim was…”).
Response 6: Thanks for your kind comment. We have changed description about what was done (“we performed…”) into the aim of the study should be presented (e.g. “The aim was…”) in the manuscript. The added sentence is “with the aim to explore whether the anti-obesity capacity of lotus leaf will enhance after fermentation with Enterococcus and try to dig out the potential and corresponding mechanism to provide a foundation of usage of lotus leaf as a dietary supplement for the prevention and attenuation of obesity” (line 115-118).
Materials and methods:
Point 1: It seems that Authors prepared only 1 batch of lotus (without any repetitions, or samples from various batches) – it should be clearly presented.
Response 1: Thanks for your thoughtful comment. To eliminate the difference between different batches of lotus leaves, we purchased 500 g lotus leaf from Taobao application at https://m.tb.cn/h.UY2KvgW?tk=TBWz2wkdERJ CZ0001, which was adequate for our research. Afterward, we prepared samples twice for in vitro and in vivo, respectively.
Point 2: The source of lotus should be described
Response 2: Thanks for your valuable comment. The lotus leaf was purchased from Taobao application and the link is https://m.tb.cn/h.UY2KvgW?tk=TBWz2wkdERJ CZ0001.
Point 3: Authors should indicate if animals were housed together or separately and how it could have influenced experienced stress.
Response 3: Thanks for your kind suggestion. The animals were housed by groups, namely eight rats were housed in a big cage that could enough for them to move and there were total six cages.
Point 4: It seems that Authors did not verify normality of distribution.
Response 4: Thanks for your thoughtful advice. We have indicated clearly in the revised manuscript that one-way analysis of variance and a Tukey’s multiple comparisons test were used for statistical analysis and p-values less than 0.05 were considered statistically significant (line 242-245). Therefore, the data were not necessary conform to normal distribution.
Point 5: Authors should verify normality of distribution and only for parametric data they should present mean and SD, while for non-parametric they should present median, min and max values.
Response 5: Thanks for your valuable suggestion. Statistical comparisons in the paper were performed using one-way ANOVA and values of p < 0.05 were considered statistically significant. We analyzed our data according to Wu (Wu et al., 2020).
Wu, Y.; Tan, F.; Zhang, T.; Xie, B.; Ran, L.; Zhao, X. The anti-obesity effect of lotus leaves on high-fat-diet-induced obesity by modulating lipid metabolism in C57BL/6J mice. Applied Biological Chemistry 2020, 63, 1-11.
Point 6: Authors should use statistical tests based on the distributions observed.
Response 6: Thanks for your thoughtful comment. We have chosen the appropriate statistical tests on the distributions observed as you kindly suggested.
Results:
Point 1: Instead of figures (charts), Authors should rather present tables, as the results are hard to follow
Response 1: Thank for your thoughtful comment. We have changed the “bar graphs” into “Scatter plot with bar” in all the graphs to see the dispersion of the data and the number of each replicates.
Discussion:
Point 1: In this section, Authors presented a number of information which do not discuss the results of the study in any way (e.g. lines 347-353, lines 358-364, etc). Instead, Authors should focus on specific results.
Response 1: Thanks for your thoughtful comment. We have revised the discussion section and focused on the specific results as well as compared our results with the other researchers (such as line 371-375 and 401-405).
Point 2: Authors should in their discussion include 3 areas: (1) compare gathered data with the results by other authors, (2) formulate implications of the results of their study and studies by other authors, (3) formulate the future areas which should be studied.
Response 2: Thanks for your thoughtful comment. We have added and revised our discussion section according to your comments to make the discussion section more profound and the added part were heighted by yellow color (such as line 371-375 and 379-389).
Point 3: Limitations of the study should be presented.
Response 3: Thanks for your thoughtful comment. We deemed that there still some limitations remained. Firstly, the effect of lotus leaf fermentation supernatant on obesity we explored were only at the phenomenon level, therefore more experiments are needed to further explore the specific mechanism. Secondly, we used SD rats as model organisms to explore the functional characteristic of FLLS-WEHI01 in vitro, which indeed could not fully reflect the effect in the human body. Therefore, further studies are needed to elucidate the biological effects of FLLS-WEHI01 on obesity-related disorders and to clarify its effect on the health of human.
Point 4: It seems that Authors prepared only 1 batch of lotus (without any repetitions, or samples from various batches) – it should be presented as a limitation of the study.
Response 4: Thanks for your thoughtful comment. To eliminate the difference between different batches of lotus leaves, we purchased 500 g lotus leaf from Taobao application at https://m.tb.cn/h.UY2KvgW?tk=TBWz2wkdERJ CZ0001, which was adequate for our research.
Point 5: Figure 9 should not be presented in the Discussion Section.
Response 5: Thanks for your kind comment. We have put Figure 9 in the conclusion section (line 446).
Conclusions:
Point 1: Authors should focus on the conclusions formulated directly based on their study.
Response 1: Thanks for your kind comment. We have revised the conclusion which directly based on our study as you suggested (line 440-450).
Authors’ contributions:
Point 1: It seems that contribution of YT, LQ, WX, XH, HW was only minor and they did not participate in preparing manuscript. There is a serious risk of a guest authorship procedure which is forbidden. In such case they should be rather presented in Acknowledgements Section and not be indicated as authors of the study.
Response 1: Thanks for your valuable suggestion. The contribution of authors can be summarized as follow: HW conceived the study and designed the experiments, YT performed the experiments and prepared the figures, and LQ, WX, and XH analyzed the data. We think all the parts are equally important for a formation of an article, so we suppose all the people should be listed as authors instead of only be presented in Acknowledgements Section.
Round 2
Reviewer 1 Report
The authors have satisfactorily answered the questions
Author Response
Thank you for your affirmation of our answer. This time, we have optimized the background, discussion and data of the article to better present the article.
Reviewer 2 Report
The manuscript entitled „Lotus (Nelumbo nucifera Gaertn.) Leaf Fermentation Supernatant Inhibits Adipogenesis in 3T3-L1 Preadipocytes and Suppresses Obesity in High-Fat Diet-Induced Obese Rats” presents interesting issue, but some major problems must be corrected.
It seems that Authors are not familiar with the scientific background of the described issue. Authors even do not know the definition of obesity, as they describe “obese people with mild symptoms of BMI < 25”, while the definition of obesity by the World Health Organization indicates that obesity is defined based on BMI and is indicated for BMI > 30 (only) (https://www.who.int/europe/news-room/fact-sheets/item/a-healthy-lifestyle---who-recommendations). This is the basic nutritional knowledge and indicating the improper information is a really major problem.
Moreover, it is not the only problem. Similarly, while indicating the recommendations of body mass reduction, Authors formulate improper information that are not based on any recommendations or guidelines.
The manuscript should not be published in the present form.
General:
The English language should be polished – it seems as Authors are not native English speakers and some sentences are hard to follow and to understand – preferable they should have their manuscript polished by a professional English correcting agency.
E.g. (please find 2 examples, but the corrections should be made in the whole text):
- instead of “available” (“LAB, is available in plenty of fermented foods, and gut of human as well as animals”), rather “present” should be used
- “Obesity arises from the imbalance of energy intake and expenditure, and the cellular development was associated with adipose tissue growth involves both cellular hypertrophy (increase in size) and hyperplasia (increase in number), where cell hypertrophy is primarily determined by adipogenesis” – Authors use at the same time past tense and present tense, which is not justified
Abstract:
Authors should present specific numeric results accompanied by the results of the statistical analysis (p-Values).
Introduction:
“Multiple therapeutic methods with which to treat obesity are available, including lifestyle intervention, formula diet, drugs as well as bariatric surgery” – Authors should define “formula diet” and should present the approaches listed as those more recommended (lifestyle and dietary interventions) and those to be applied if the other are not effective only (bariatric surgery, pharmacotherapy)
Authors should present a proper approach based o the recommendations
The functional foods other than lotus should be at least listed and their action should be described
Specific mechanisms of lotus actions should be described
Last paragraph is messy and it presents some random studies not associated with each other – Authors should present only necessary information
Materials and methods:
It seems that Authors prepared only 1 batch of lotus (without any repetitions, or samples from various batches) – it should be clearly presented
The source of lotus should be described – place where it was planted, season, conditions (not the seller)
It seems that Authors did not verify normality of distribution – it should be indicated in Statistical Analysis Sub-section.
Authors should verify normality of distribution and only for parametric data they should present mean and SD, while for non-parametric they should present median, min and max values.
Authors should use statistical tests based on the distributions observed.
Results:
It seems that Authors did not verify normality of distribution.
Authors should verify normality of distribution and only for parametric data they should present mean and SD, while for non-parametric they should present median, min and max values.
Authors should use statistical tests based on the distributions observed.
Instead of figures (charts), Authors should rather present tables, as the results are hard to follow
Discussion:
In this section, Authors presented a number of information which do not discuss the results of the study in any way. Instead, Authors should focus on specific results.
Authors should in their discussion include 3 areas: (1) compare gathered data with the results by other authors, (2) formulate implications of the results of their study and studies by other authors, (3) formulate the future areas which should be studied.
Limitations of the study should be presented.
It seems that Authors prepared only 1 batch of lotus (without any repetitions, or samples from various batches) – it should be presented as a limitation of the study.
Conclusions:
Authors should focus on the conclusions formulated directly based on their study.
Figure 9 should not be presented in the Conclusions Section.
Authors’ contributions:
It seems that contribution of YT, LQ, WX, XH, HW was only minor and they did not participate in preparing manuscript. There is a serious risk of a guest authorship procedure which is forbidden. In such case they should be rather presented in Acknowledgements Section and not be indicated as authors of the study.
Author Response
Dear Editors and Reviewers:
Thanks for your letter and for the reviewers’ comments concerning our manuscript, entitled “Lotus (Nelumbo nucifera Gaertn.) Leaf Fermentation Supernatant Inhibits Adipogenesis in 3T3-L1 Preadipocytes and Suppresses Obesity in High-Fat Diet-Induced Obese Rats” (ID: nutrients-1911079). Those comments are all valuable and very helpful for revising and improving our paper, as well as the important guiding significance to our research. We have comprehensively considered your opinions and tried our best to revise the manuscript according to the Reviewers’ pertinent comments. Revised portions are marked in yellow in the renew manuscript. We hope that the revision will meet with your requirements and looking forward to your positive decision.
Yours sincerely
Dr. Xueying Tao
State Key Laboratory of Food Science and Technology, Nanchang University, Nanchang 330047, P. R. China.
Tel: +86-791-88334578;
Fax: +86-791-8833-3708;
E-mail: [email protected]
The following is a point-to-point response to the reviewers’ comments:
Author’s response to reviewer’s comments
General:
Point 1: The English language should be polished – it seems as Authors are not native English speakers and some sentences are hard to follow and to understand – preferable they should have their manuscript polished by a professional English correcting agency.
E.g. (please find 2 examples, but the corrections should be made in the whole text):
- instead of “available” (“LAB, is available in plenty of fermented foods, and gut of human as well as animals”), rather “present” should be used.
- “Obesity arises from the imbalance of energy intake and expenditure, and the cellular development was associated with adipose tissue growth involves both cellular hypertrophy (increase in size) and hyperplasia (increase in number), where cell hypertrophy is primarily determined by adipogenesis” – Authors use at the same time past tense and present tense, which is not justified.
Response1: Thanks for your valuable suggestion. Firstly, we have corrected the improper statements including the two examples you have given in the comment, and carefully checked and improved the English writing in the revised manuscript. Further, we have also sent our manuscript to polish at https://www.mdpi.com/authors/english and the English Editing ID is english-51400. We hope that the revised manuscript can meet the professional standards.
Abstract:
Point1: Authors should present specific numeric results accompanied by the results of the statistical analysis (p-Values).
Response1: Thanks for your valuable comment. We have presented the specific numeric results accompanied by the results of the statistical analysis (p-Values) (exception occur when p value is less than 0.001 because the specific numeric results can’t show under this circumstance) as you kindly suggested, which mainly focused on line 30-41.
Introduction:
Point 1: “Multiple therapeutic methods with which to treat obesity are available, including lifestyle intervention, formula diet, drugs as well as bariatric surgery” – Authors should define “formula diet” and should present the approaches listed as those more recommended (lifestyle and dietary interventions) and those to be applied if the other are not effective only (bariatric surgery, pharmacotherapy)
Response 1: Thanks for your constructive comment. We have corrected it according to the recommendations and added the information about the definition of “formula diet” as well as the approaches of lifestyle, dietary interventions, bariatric surgery and pharmacotherapy, which was presented in line 61-79.
Point 2: Authors should present a proper approach based on the recommendations
Response 2: Sorry for giving improper information about some recommendations of body mass reduction. We have corrected and presented the proper approaches and information based on the recommendations, guidelines and references.
Point 3: The functional foods other than lotus should be at least listed and their action should be described
Response 3: Thanks for your valuable advice. As you kindly suggested, we have listed some functional foods containing tea (green tea, dark tea and oolong tea), the Radix Pueraria lobata as well as Lactobacillus plantarum BL2 fermented garlic extract and described their action on anti-obesity, as presented in line 84-98.
Point 4: Specific mechanisms of lotus actions should be described
Response 4: Thanks for your thoughtful comment. We have added the specific mechanisms of lotus action on anti-obesity by providing an example in detail as well as summarizing a general mechanism (amelioration of intestinal integrity, relief of chronic inflammation, and modulation of the gut microflora via specific metabolites such as short chain fatty acids.), as presented in line 113-121.
Point 5: Last paragraph is messy and it presents some random studies not associated with each other – Authors should present only necessary information
Response 5: Thanks for your thoughtful comment. We have deleted the messy and non-associated information and only presented the necessary information which contained the backgrounds, preliminary studies and aim of this research (line 126-138).
Materials and methods:
Point1: It seems that Authors prepared only 1 batch of lotus (without any repetitions, or samples from various batches) – it should be clearly presented.
Response 1: Thanks for your constructive comment. We designed three experiments to explore the anti-obesity of Enterococcus fermented lotus leaf. First, we compared the differences of bioactive substances between fermented lotus leaf and unfermented (the related data were not shown in the paper), then we tried to clarify the anti-obesity effect in vitro by 3T3 preadipocyte and the finally we elaborated the inhibitive effect in vivo by high fat diet-induced obese rats. There actually included more than one performance in each experiment, that’s to say we prepared the samples for many times and the results kept the stable within samples prepared in different batches, which indicated that the Enterococcus fermented lotus leaf possess the capacity to inhibit obesity.
Point 2: The source of lotus should be described – place where it was planted, season, conditions (not the seller).
Response 2: Thanks for your valuable advice. The lotus leaves we used in the experiments grow naturally at the end of June and early July each year in the Weishan Lake, which is a fault lake located in the south of Weishan County, Jining City, Shandong Province, China. It’s known that Weishan Lake is a protective water source area cause it’s an important area for fishermen, which could ensure the lotus leaves natural and pollution-free. And the corresponding information were also added in the revised manuscript in line 148-152.
Point 3: It seems that Authors did not verify normality of distribution – it should be indicated in Statistical Analysis Sub-section.
Response 3: Thanks for your kind suggestion. We have indicated clearly in Statistical Analysis Section and it was presented as follow “The data and the normal distribution of the data (p of Shapiro-Wilk normality greater than 0.05 were thought conformed to normal distribution) were analyzed using GraphPad Prism 7 statistical software. The results were expressed as the mean ± standard deviation (SD). All the data passed the verification of normal distribution were analyzed via one-way analysis of variance and Tukey’s multiple comparisons test were used for comparisons between-groups, while the others were analyzed via non-parametric test. The p-values less than 0.05 were considered statistically significant.” (Line 248-255)
Point 4: Authors should verify normality of distribution and only for parametric data they should present mean and SD, while for non-parametric they should present median, min and max values.
Response 4: Thanks for your kind suggestion. We have indicated clearly in Statistical Analysis Section that we have verified of normality of distribution of the data and selected the suitable statistical analysis based on the results, namely the data that didn’t pass the verification of normality of distribution were analysed by non-parametric test.
Point 5: Authors should use statistical tests based on the distributions observed.
Response 5: Thanks for your valuable comment. We have checked carefully whether our data passed the verification of normality of distribution, and have selected the suitable statistical tests based on the results.
Point 2: Authors should verify normality of distribution and only for parametric data they should present mean and SD, while for non-parametric they should present median, min and max values.
Response 2: Thanks for your valuable suggestion. We have verified the data about the normality of distribution according to your kind comment and some normality distribution maps and corresponding parameters are showed as Fig S1. Of course, some of our data are not passed the Gaussian distribution (namely Fig 3a, 5b, 6a, 6b, 6f), which were re-analyzed via non-parametric test and the figures as well as the descriptions about them were all corrected in the new uploaded manuscript. Thanks again for reminding us to select the correct statistic test, which are beneficial not only to this study but also our future works.
Fig S1. The normal distribution map and corresponding parameters of some data in our study. (a) The normal distribution map of Fig 1; (b) Fig 2; (c) Fig 3d; (d) Fig 4b; (e) Fig 5c; (f) Fig 5d; (g) Fig 6c; (h) epididymis fat weight.
Point 3: Authors should use statistical tests based on the distributions observed.
Response 3: Thanks for your valuable comment. We have checked carefully whether our data passed the normality of distribution, and have chosen the suitable statistical tests based on the results.
Point 4: Instead of figures (charts), Authors should rather present tables, as the results are hard to follow
Response 4: Thank for your thoughtful comment. Firstly, we have changed the “bar graphs” into “Scatter plot with bar” in all the graphs to see the dispersion of the data and the number of each replicates. Besides, we changed the liver, spleen, kidney, and epididymis fat weight as well as food intake of rats from figure into table (namely table 1) to make the data easier to follow as you kindly advised.
Discussion:
Point 1: In this section, Authors presented a number of information which do not discuss the results of the study in any way. Instead, Authors should focus on specific results.
Response 1: Thanks for your constructive suggestion. We have deleted some information in the Discussion Section according to your comment, such as the opening part of the second paragraph which were more suitable presented in Introduction Section. Besides, we focused more on specific results and tried to discuss them more by comparing with other authors’ studies according to your comment.
Point 2: Authors should in their discussion include 3 areas: (1) compare gathered data with the results by other authors, (2) formulate implications of the results of their study and studies by other authors, (3) formulate the future areas which should be studied.
Response 2: Thanks for your thoughtful comment. We have added and revised our discussion section according to your comments to make the discussion section more profound and the added parts were heighted by yellow colour (such as line 372-390, 401-409 and 442-451). In vitro assay, our discussion focused on specific results of our study, and compared them with other’s work with the similar or converse results. Contrary, we give a general discussion about the results of experiment in vivo, and spent longer length to discuss the relations between the parameters we measured to make the study easier to understand. Of course, we also cited some works of other researchers and compared them with ours, thus to deduce where the limitations our study lay and to absorb the important guiding significance to our research.
Point 3: Limitations of the study should be presented.
Response 3: Thanks for your thoughtful comment. We deemed that there still some limitations remain. Firstly, the effect of lotus leaf fermentation supernatant on obesity we explored were only at the phenomenon level, therefore more experiments are needed to further explore the specific mechanism. Secondly, we used SD rats as model organisms to explore the functional characteristic of FLLS-WEHI01 in vitro, which indeed could not fully reflect the effect in the human body. Therefore, further studies are needed to elucidate the biological effects of FLLS-WEHI01 on obesity-related disorders and to clarify its effect on the health of human. And the corresponding information were also added in the revised manuscript in line 442-451.
Point 4: It seems that Authors prepared only 1 batch of lotus (without any repetitions, or samples from various batches) – it should be presented as a limitation of the study.
Response 4: Thanks for your constructive suggestion. We have clarified clearly about the preparations of experimental samples in the point 1 in Materials and methods, and we prepared the samples for more than three times and the results kept the stable within samples prepared in different batches, which indicated that the Enterococcus fermented lotus leaf possess the capacity to inhibit obesity.
Conclusions:
Point 1: Authors should focus on the conclusions formulated directly based on their study.
Response 1: Thanks for your kind comment. We have revised the conclusion which directly based on our study as you suggested and now the conclusion turned into “The oral administration of FLLS-WEHI01 significantly decreased body weight, epididymis fat mass, TC and TG in liver, LDL-C levels, and fasting blood glucose, in addition to improving insulin resistance in HFD-induced obese rats, which might have occurred through the modulation of adipogenic transcription factors such as Pparγ, Cebpα, Acc and Fas. Therefore, the results proved that FLLS-WEHI01 could potentially be applied as a dietary supplement for the prevention and attenuation of obesity.” (line 454-460).
Point 2: Figure 9 should not be presented in the Conclusions Section.
Response 2: Thanks for your valuable advice. We have deleted Figure 9 from the Conclusion Section and presented it only as a graphic summary to reflect the anti-obesity effect of the supernatant fermented lotus leaf with Enterococcus.
Authors’ contributions:
Point 1: It seems that contribution of YT, LQ, WX, XH, HW was only minor and they did not participate in preparing manuscript. There is a serious risk of a guest authorship procedure which is forbidden. In such case they should be rather presented in Acknowledgements Section and not be indicated as authors of the study.
Response 1: Thanks for your valuable suggestion. The contribution of authors can be summarized as follow: XT and HW conceived the study and designed the experiments, YT performed the experiments, LQ, WX, and XH analyzed the data, YH and YT prepared the figures and wrote the draft, and XT interpreted the results and revised the manuscript. We think all the parts are equally important for a formation of an article, so we suppose all the people should be listed as authors instead of only be presented in Acknowledgements Section.